# Two point mutations in protocadherin-1 disrupt hantavirus recognition and afford protection against lethal infection

Megan M. Slough [1,9], Rong Li[2,9], Andrew S. Herbert[3,9], Gorka Lasso[1], Ana I. Kuehne[3], Stephanie R. Monticelli[3,4], Russell R. Bakken[3], Yanan Liu[2], Agnidipta Ghosh[5], Alicia M. Moreau[3], Xiankun Zeng [3], Félix A. Rey [6], Pablo Guardado-Calvo [6,7], Steven C. Almo[5], John M. Dye[3], Rohit K. Jangra [1,8,10] ✉, Zhongde Wang [2,10] ✉ & Kartik Chandran [1,10] ✉

Andes virus (ANDV) and Sin Nombre virus (SNV) are the etiologic agents of severe hantavirus cardiopulmonary syndrome (HCPS) in the Americas for which no FDA-approved countermeasures are available. Protocadherin-1 (PCDH1), a cadherin-superfamily protein recently identified as a critical host factor for ANDV and SNV, represents a new antiviral target; however, its precise role remains to be elucidated. Here, we use computational and experimental approaches to delineate the binding surface of the hantavirus glycoprotein complex on PCDH1's first extracellular cadherin repeat domain. Strikingly, a single amino acid residue in this PCDH1 surface influences the host species-specificity of SNV glycoprotein-PCDH1 interaction and cell entry. Mutation of this and a neighboring residue substantially protects Syrian hamsters from pulmonary disease and death caused by ANDV. We conclude that PCDH1 is a bona fide entry receptor for ANDV and SNV whose direct interaction with hantavirus glycoproteins could be targeted to develop new interventions against HCPS.

Rodent-borne orthohantaviruses (hereafter, hantaviruses) are segmented, negative-strand RNA viruses that have co-evolved with their rodent hosts over millions of years, by some estimates[1]. Zoonotic transmission of some hantaviruses can cause two diseases in humans, hemorrhagic fever with renal syndrome (HFRS) and hantavirus cardiopulmonary syndrome (HCPS)[2]. Both cause significant morbidity and mortality with case-fatality rates approaching 15% (HFRS) and 40% (HCPS)[3]. No FDA-approved specific antivirals are available to treat HCPS or HFRS, and their

development is challenged by the limited understanding of the hantavirus life cycle.

Hantaviruses belonging to the 'New World' clade, such as Andes virus (ANDV) and Sin Nombre virus (SNV), are the major etiologic agents of HCPS in South and North America, respectively[2]. 'Old World' hantaviruses found primarily in Asia and Europe, such as Hantaan virus (HTNV) and Seoul virus (SEOV), can cause HFRS in humans[2]. We recently identified protocadherin-1 (PCDH1), a member of the non-clustered protocadherins in the cadherin superfamily[4,5], as a clade-

[1]Department of Microbiology and Immunology, Albert Einstein College of Medicine, Bronx, NY, USA. [2]Department of Animal, Dairy and Veterinary Sciences, Utah State University, Logan, UT, USA. [3]United States Army Medical Research Institute of Infectious Diseases, Fort Detrick, MD, USA. [4]The Geneva Foundation, Tacoma, WA, USA. [5]Department of Biochemistry, Albert Einstein College of Medicine, Bronx, NY, USA. [6]Institut Pasteur, Université Paris Cité, CNRS UMR3569, Structural Virology Unit, F-75015 Paris, France. [7]Institut Pasteur, Université Paris Cité, Structural Biology of Infectious Diseases Unit, F-75015 Paris, France. [8]Present address: Microbiology and Immunology, Louisiana State University Health Sciences Center-Shreveport, Shreveport, LA, USA. [9]These authors contributed equally: Megan M. Slough, Rong Li, Andrew S. Herbert. [10]These authors jointly supervised this work: Rohit K. Jangra, Zhongde Wang, Kartik Chandran. ✉e-mail: rohit.jangra@lsuhs.edu; zonda.wang@usu.edu; kartik.chandran@einsteinmed.edu

specific entry host factor for New World hantaviruses. PCDH1 directly engages the tetrameric viral Gn/Gc glycoprotein complex[6,7], which mediates cell entry[8]. PCDH1 is expressed in the airway epithelium, where it colocalizes with E-cadherin at apical cell-cell contact sites[9–11], and in vascular endothelial cells, the primary targets for hantavirus infection[12]. *PCDH1* is a susceptibility gene for airway hyper-responsiveness and asthma[11,13,14], and its gene product appears to regulate airway epithelial barrier function through mechanisms that remain unclear[10].

PCDH1 is a Type I transmembrane protein comprising seven extracellular cadherin (EC) repeat domains, a 'protocadherin domain' that encompasses juxtamembrane and transmembrane sequences, and a long cytoplasmic tail[15,16]. A soluble PCDH1 fragment, consisting of the first four EC repeat domains (EC1–4), crystallized as a head-to-tail homodimer making critical contacts between EC1 and EC4[17]. This suggests a mechanism by which PCDH1 molecules on neighboring cells form adhesive contacts[17]. As reported previously, PCDH1's EC1 is necessary and sufficient for binding by New World hantavirus Gn/Gc, and PCDH1 loss greatly reduces the lethality of ANDV infection in a Syrian hamster model of HCPS[6]. However, the precise contacts at the PCDH1:Gn/Gc interface and the effects of PCDH1:Gn/Gc interaction on PCDH1 oligomerization (and vice versa) remain undefined.

Herein, we link the inter-species sequence variations in PCDH1 and mammalian host-specific receptor usage by hantaviruses to pinpoint sequences that could influence susceptibility to hantavirus infection. We then build on this information and combine structure-based prediction, comprehensive site-directed mutagenesis, and protein engineering to map Gn/Gc's binding site in PCDH1 and shed light on the mode of hantavirus-PCDH1 recognition during viral entry.

## Results

### Murine cells are less susceptible than human cells to SNV entry

Natural infection by rodent-borne hantaviruses is proposed to be largely host-specific, suggesting the existence of molecular barriers to cross-species viral infection[3,18,19]. Specifically, ANDV and SNV, whose reservoir hosts are the long-tailed pygmy rice rat (*Oryzomys longicaudatus*) and deer mouse (*Peromyscus maniculatus*), respectively, appear unable to infect the house mouse (*Mus musculus*) as no successful experimental infections of the latter have been reported. Further, mice engineered to lack a functional type I interferon response were reported to be highly susceptible to HTNV[20] and SEOV infection[21], but have not been shown to support ANDV or SNV infection. Thus, we postulated that there may be murine barriers to ANDV and SNV infection at the cellular level. To test this hypothesis, we infected primary human pulmonary microvascular endothelial cells (HPMECs) and primary murine lung microvascular endothelial cells (MLMECs) with replication-competent, recombinant vesicular stomatitis viruses (rVSVs) expressing ANDV (rVSV-ANDV-Gn/Gc) or SNV (rVSV-SNV-Gn/Gc) Gn/Gc. Mouse cells showed greatly (~100-fold) reduced susceptibility to rVSV-SNV-Gn/Gc and slightly (~2-fold) reduced susceptibility to rVSV-ANDV-Gn/Gc relative to the human cells (Fig. 1a).

### Residue 83 influences susceptibility and Gn/Gc recognition

We postulated that differences in PCDH1 expression levels and/or sequence could account for the human-murine difference in viral entry into primary lung endothelial cells. However, immunostaining indicated that PCDH1 was abundantly expressed at the cell surface in mouse endothelial cells (Fig. 1b). We found, instead, that ectopic expression of human PCDH1 in MLMECs enhanced infection of rVSV-SNV-Gn/Gc (Fig. 1c–d), indicating that a molecular incompatibility between SNV Gn/Gc and murine PCDH1 is at least partially responsible for the entry block in murine endothelial cells.

We next examined the possibility that the murine ortholog of PCDH1 harbors amino acids that reduce its function as a hantavirus entry factor. Indeed, alignment of the EC1 domain sequences in PCDH1

from humans, non-human primates, and rodents revealed human-mouse sequence differences at three positions: F83L, L127I, and D130N (Supplementary Data 1 and Fig. 1e). To evaluate the effect of these EC1 residues on PCDH1's receptor activity, we ectopically expressed EC1-'murinized' variants of human PCDH1 in *PCDH1*-knockout (KO) human osteosarcoma U2OS cells and tested their susceptibility to both, rVSVs bearing ANDV or SNV Gn/Gc, and the authentic agents. Although all the PCDH1 variants expressed well and localized to the cell surface (Supplementary Fig. 1), only the F83L substitution failed to restore cellular susceptibility to SNV entry and infection, suggesting F83L is determinative (Fig. 1f). Moreover, over-expression of human PCDH1 bearing F83L failed to enhance rVSV-SNV-Gn/Gc infection in MLMECs (Fig. 1c–d), supporting the conclusion that the human-murine sequence difference at PCDH1 position 83 renders murine endothelial cells less susceptible to SNV Gn/Gc-dependent entry. As shown previously, PCDH1 was dispensable for viral infection mediated by an Old-World hantavirus Gn/Gc (HTNV)[6,7] (Fig. 1f).

Finally, we assessed the Gn/Gc binding activities of soluble, human PCDH1 comprising its first two EC domains (sEC1-2)[6] and bearing either the human or murine residue at position 83 (WT or F83L, respectively). We initially attempted binding studies with a recombinant soluble Gn/Gc monomer[22] but could not detect sEC1-2 binding, suggesting authentic Gn/Gc tetramers are required for the Gn/Gc:EC1 interaction (Supplementary Fig. 2). Because soluble tetramers have not yet been described, we carried out three distinct binding assays with viral particles displaying Gn/Gc tetramers. First, we measured the capture of rVSV particles bearing ANDV or SNV Gn/Gc (Supplementary Fig. 3) onto sEC1-2–coated ELISA plates (Fig. 2a). Binding of rVSV-SNV-Gn/Gc to sEC-2 was abrogated by the F83L mutation (Fig. 2b), consistent with our findings with SNV in MLMECs (Fig. 1a–c). By contrast, rVSV-ANDV-Gn/Gc bound equivalently to both sEC1-2(WT) and sEC-2(F38L) (Fig. 2b). However, it remained possible that the inherently avid interactions measured in this assay could mask putative small or moderate reductions in ANDV Gn/Gc:PCDH1 affinity. To account for this, we performed a competition ELISA, in which rVSV-ANDV-Gn/Gc and rVSV-SNV-Gn/Gc particles were incubated with WT or F83L sEC1-2 in solution, and particles with receptor-free Gn/Gc sites were then captured onto WT sEC1-2–coated plates (Fig. 2c). Pre-incubation with sEC1-2(WT) inhibited rVSV-ANDV-Gn/Gc capture in a sEC1-2 dose-dependent manner (Fig. 2d), whereas sEC1-2(F83L) displayed substantially weaker, though still detectable blocking activity. Similar results were obtained in a third assay in which viral particles were pre-incubated with sEC1-2 in solution as above, and then exposed to PCDH1-bearing target cells (Fig. 2e): both sEC1-2(WT) and sEC1-2(F83L) could block rVSV-ANDV-Gn/Gc infection, but the latter was less active (Fig. 2f). The lack of inhibition of rVSV-SNV-Gn/Gc by sEC1-2(F83L) but not sEC1-2(WT) in these follow-up assays (Fig. 2c, d) was concordant with the essentially complete loss in its binding to sEC1-2(F83L) observed in the direct binding ELISA (Fig. 2a, b). Together, these findings indicate that the F83L mutation abolishes PCDH1 binding to SNV Gn/Gc, explaining the resistance of MLMECs to SNV entry and infection and the enhancing effect of human PCDH1 expression in these cells. We surmise that avid interactions between ANDV Gn/Gc and murine PCDH1 on cell surfaces can compensate for their reduced binding affinity, just as we observed in the avid binding assays with sEC1-2(F83L) (Fig. 2a, b).

### Structure-based prediction of PCDH1:Gn/Gc interfacial residues

We postulated that F83 is a key contact residue within a larger Gn/Gc-binding surface in PCDH1. F83 is located in a disordered, membrane-distal loop in EC1 that is not resolved in the crystal structure of PCDH1 (encompassing EC1-4, PDB 6MGA)[17]. We modeled multiple conformations of the missing loop, ranging from a more "closed" (buried) to a more "open" (solvent-accessible) conformation, which likely reflect the loop's intrinsic flexibility (Fig. 3a). To identify F83 neighboring

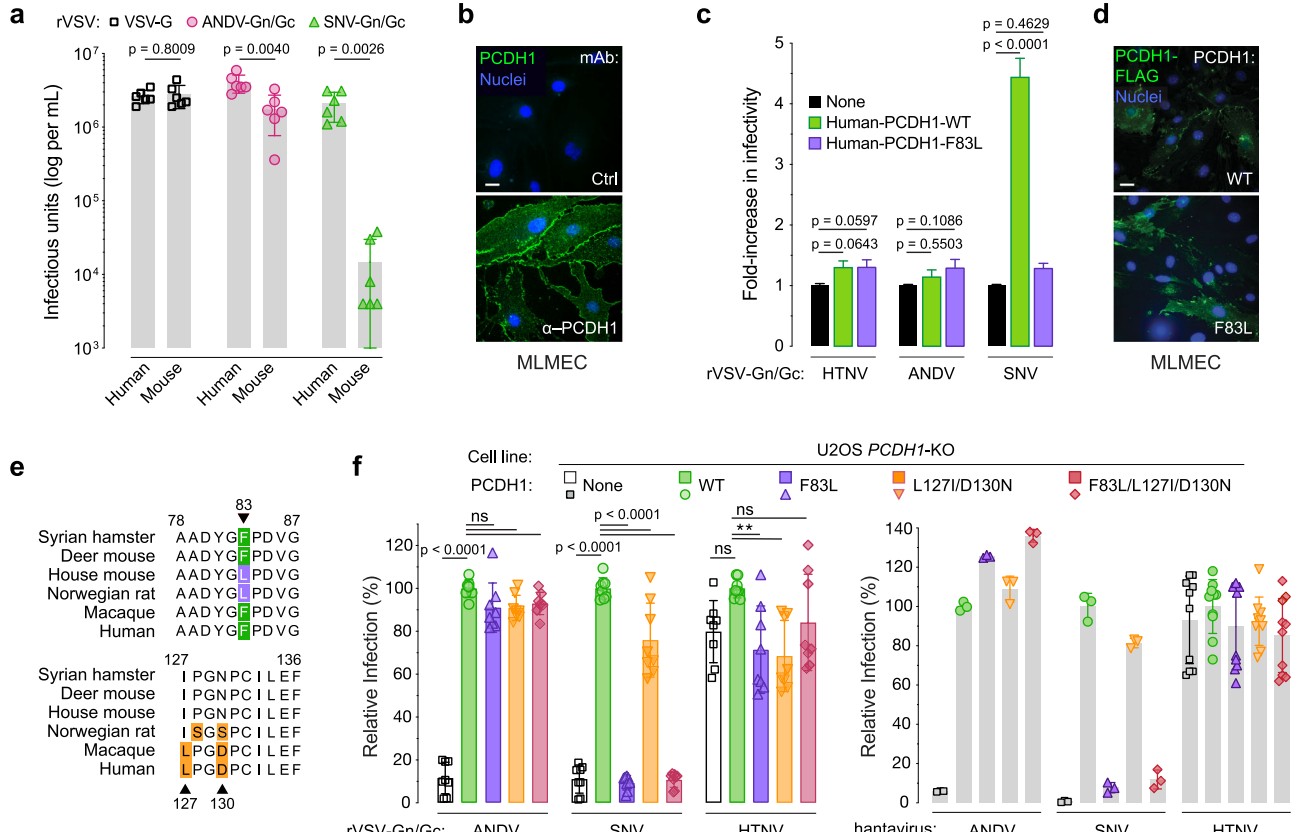

**Fig. 1 | Residue F83 in PCDH1 is a key determinant of Sin Nombre virus infection. a** Viral titer of rVSVs expressing G, ANDV Gn/Gc, or SNV Gn/Gc in human or mouse primary lung endothelial cells. Means ± SD: $n = 6$ wells of infected cells examined over three independent experiments. **b** Surface expression of endogenous PCDH1 in primary mouse lung microvascular endothelial cells (MLMECs) used in (**a**). Cells were immunostained with PCDH1-specific monoclonal antibody (mAb) 3305 or a negative control mAb (Ctrl.) Scale bar, 20 μm. **c** Infectivity of rVSVs bearing HTNV, ANDV, or SNV Gn/Gc in primary MLMECs expressing flag-tagged, wild-type (WT) or mouse-variant (F83L) human PCDH1. rVSV infectivities are expressed as fold change relative to that in non-complemented cells (set to one). Means ± SD: $n = 35$ wells of infected cells examined over three independent experiments. **d** Cells described in (**c**) were immunostained with an anti-flag antibody to detect total human PCDH1 expression in transfected MLMECs. Scale bar, 20 μm. **e** Alignment of human PCDH1-EC1 amino acid sequences with a selection of rodent and primate species. All of the residues within EC1 which deviate from the consensus are shown and highlighted. Residues that specifically deviate from human EC1 are indicated [green, experimentally tested SNV-susceptible hosts; purple, experimentally unknown; orange, residues in EC1 that are different between human and other species with no known link to susceptibility]. Alignments generated by Clustal Omega. **f** The capacity of U2OS *PCDH1*-KO cells expressing the indicated PCDH1 variants to support hantavirus Gn/Gc-dependent entry. Cells were exposed to rVSVs bearing the indicated Gn/Gc proteins or authentic ANDV, SNV, or HTNV. "None" indicates no PCDH1 expression. The infectivity of each virus was normalized to that obtained in U2OS *PCDH1*-KO cells complemented with WT PCDH1. rVSV means ± SD: $n = 8$ wells of infected cells examined over three independent experiments. ANDV/SNV/HTNV means ± SD: one experiment examining $n = 3$ (ANDV and SNV) or $n = 10$ (HTNV) wells of infected cells. Infectious units (**a**) were compared by unpaired, two-tailed $t$ test with Welch's correction. Infectivities (**c**, **f**) were compared by one-way ANOVA with Dunnett's correction for multiple comparisons; *ns* > 0.05, **$P$ < 0.01. Source data are provided as a Source Data file.

residues that might also mediate Gn/Gc binding, we performed structure-based, interfacial prediction using the two most divergent modeled loop conformations, in which the F83 sidechain is either mostly buried or exposed to the solvent. We ran five complimentary algorithms to predict interfacial residues in the PCDH1 ectodomain and ranked predictions according to the number of supporting algorithms. The top 18 residues, predicted by at least four methods, were localized in the same face of the EC1 domain and included F83 (Supplementary Data 2 and Fig. 3b, c). These residues were selected for further evaluation, and to capture additional, potential Gn/Gc-interacting residues that ranked lower in our analysis, we included 11 neighboring residues in our experimental studies of PCDH1:Gn/Gc binding (Fig. 3b, c).

## Experimental mapping of the ANDV Gn/Gc:PCDH1 binding interface

To assess the binding capacity of computationally predicted interacting residues in PCDH1, we mutated the 29 selected EC1 residues to alanine (A) and/or residues with reversed charges or polarity (e.g.,

lysine (K) to aspartic acid (D) and alanine to serine (S); Fig. 3c). sEC1-2 proteins bearing these point mutations were efficiently produced and largely exhibited similar electrophoretic mobilities (Supplementary Fig. 4a). Although three variants migrated anomalously during SDS-polyacrylamide gel electrophoresis (I140A, D142A, and R150A), sEC1-2(I140A) eluted as a monodisperse peak with an apparent molecular weight resembling WT sEC1-2 (~25–29 K) in a size-exclusion column[6], indicating that it is largely monomeric in solution (Supplementary Fig. 4b).

The purified sEC1-2 mutants were screened for binding to ANDV Gn/Gc by competitive ELISA (Fig. 4a). We chose to evaluate ANDV and not SNV in these screens because the weaker binding affinity of the latter Gn/Gc for PCDH1 necessitated prohibitive quantities of viral material for binding assays and was expected to reduce our capacity to detect nuanced reductions in binding by mutant sEC1-2 proteins. The sEC1-2 mutants displayed a range of effects in the competition assay (Fig. 4b). Hierarchical clustering of the competition binding curves yielded two major clades separating the highly competitive mutants (where PCDH1 binding remains unaffected; 'WT-like binders') from the

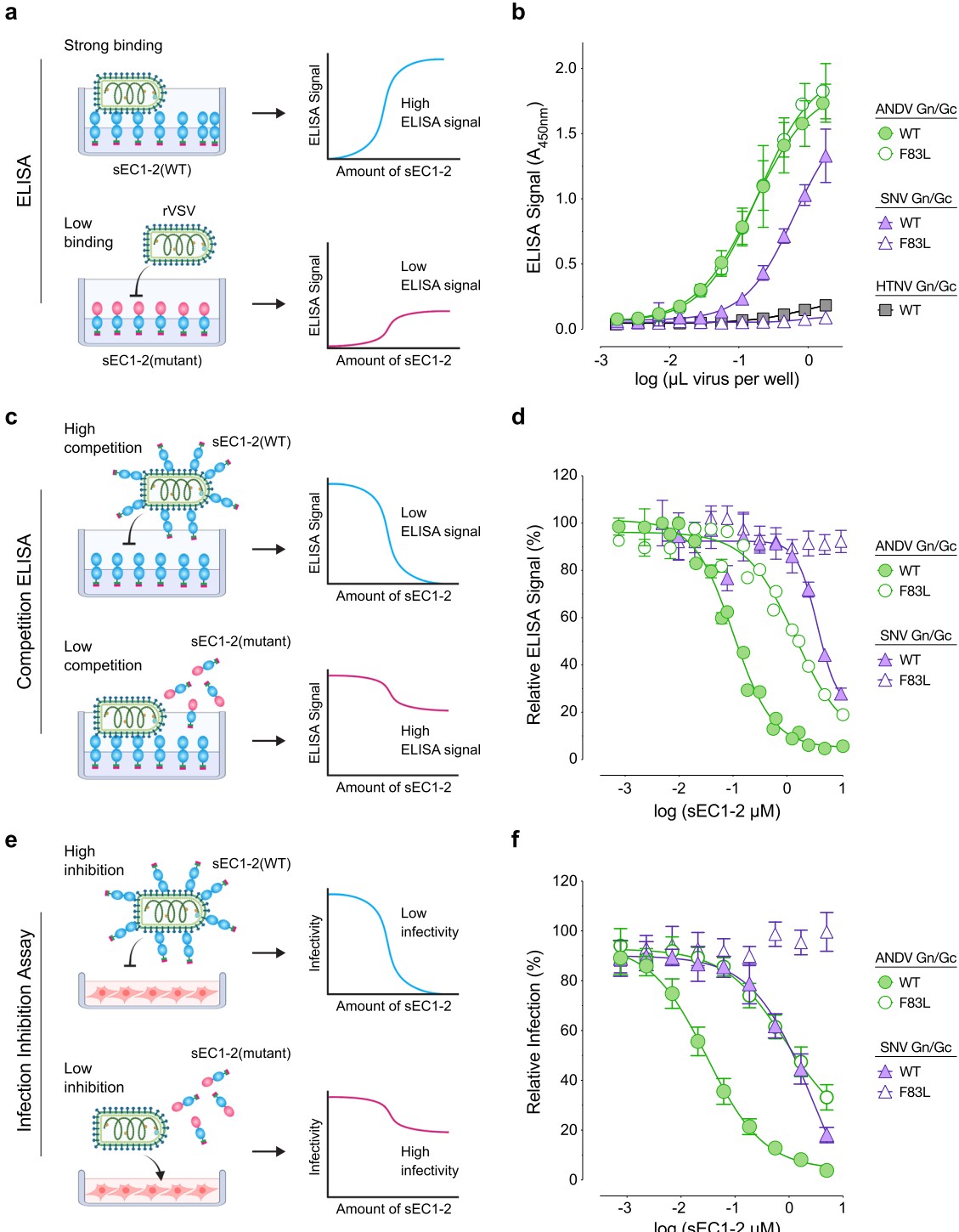

**Fig. 2 | Residue F83 in PCDH1 is critical for Sin Nombre virus Gn/Gc: sEC1-2 binding. a** Diagram of direct binding ELISA comparing sEC1-2(WT) and sEC1-2(F83L) capture of rVSVs. **b** Direct binding ELISA. rVSVs expressing ANDV, SNV, or HTNV Gn/Gc were added to sEC1-2(WT) or sEC1-2(F83L) coated ELISA plates. Means ± SD: *n* = 4 wells examined over two independent experiments. A, absorbance. **c** Diagram depicting competition ELISA comparing sEC1-2(WT) and sEC1-2(F83L) as competitive reagents. **d** Competition ELISA. rVSVs expressing ANDV or SNV Gn/Gc were pre-incubated with sEC1-2(WT) or sEC1-2(F83L) before added to sEC1-2(WT) coated ELISA plates. The ELISA signal was normalized to that obtained without competing sEC1-2. Means ± SEM: *n* = 7 wells examined over three

independent experiments (ANDV), *n* = 4 wells examined over two independent experiments (SNV). **e** Diagram depicting infection-inhibition assay comparing sEC1-2(WT) and sEC1-2(F83L) as inhibiting reagents. **f** Infection-inhibition assay using sEC1-2(WT) and sEC1-2(F83L) to block infection (MOI of 0.1) of rVSVs bearing ANDV or SNV Gn/Gc on primary human endothelial cells (HUVECs). The infectivity of each virus was normalized to that obtained without sEC1-2. Averages ± SD: *n* = 6 wells examined over two independent experiments. (sEC1-2, soluble extracellular cadherin domains 1 and 2). Figures (**a**, **c**, **e**) were created with BioRender.com. Source data are provided as a Source Data file.

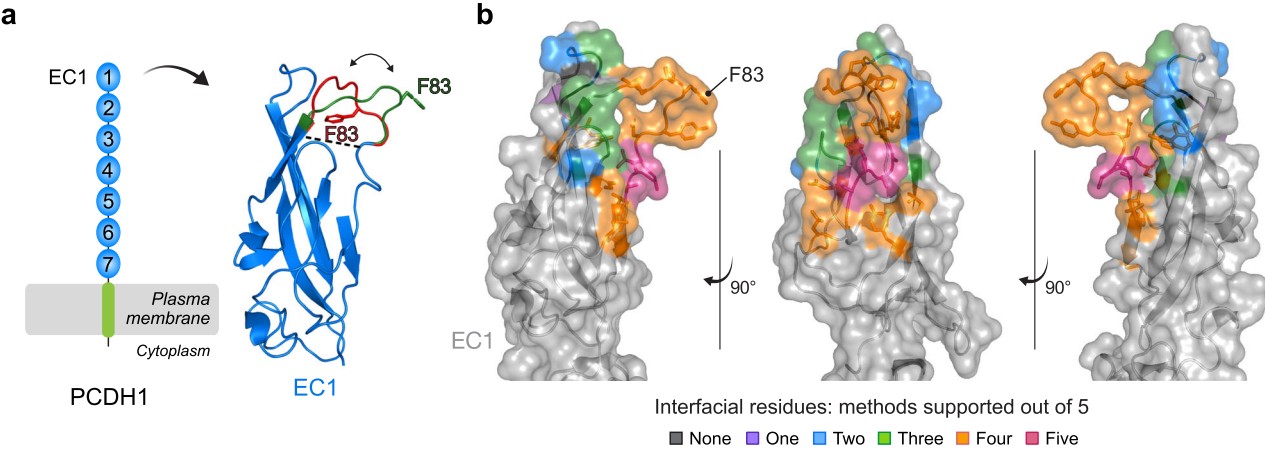

Interfacial residues: methods supported out of 5
■ None ■ One ■ Two ■ Three ■ Four ■ Five

**c**

| Residue Number | Residue Identity | Interface Prediction | Residue Change | Residue Number | Residue Identity | Interface Prediction | Residue Change |
|---|---|---|---|---|---|---|---|
| 76 | Ser | 5 | Ala | 90 | Tyr | 4 | Ala |
| 77 | Leu | 5 | Ala | 107 | Asp | 4 | Arg |
| 78 | Ala | 5 | Arg | 140 | Ile | 4 | Ala |
| 79 | Ala | 5 | Ser | 88 | His | 3 | Ala |
| 60 | Val | 4 | Ala | 104 | Lys | 3 | Ala, Glu |
| 62 | Tyr | 4 | Ala | 105 | Thr | 3 | Ala |
| 73 | Leu | 4 | Ala | 144 | Val | 3 | Ala |
| 75 | Gly | 4 | Arg | 152 | Leu | 3 | Ala |
| 80 | Asp | 4 | Ala | 102 | Asp | 2 | Ala, Arg |
| 81 | Tyr | 4 | Ala | 143 | Leu | 2 | Ala |
| 82 | Gly | 4 | Arg | 145 | Gln | 2 | Ala |
| 83 | Phe | 4 | Ala, Arg, Leu | 150 | Arg | 2 | Glu |
| 84 | Pro | 4 | Ala | 141 | Thr | 1 | Ala |
| 85 | Asp | 4 | Ala, Arg | 142 | Asp | 0 | Ala, Arg |
| 86 | Val | 4 | Ala | | | | |

**Fig. 3 | Structure-based interfacial prediction reveals a surface patch on PCDH1 EC1 that potentially drives the interaction with ANDV and SNV Gn/Gc.**
**a** Schematic representation of PCDH1 and crystal structure of EC1 (PDB 6MGA) displaying two modeled conformations (green: "open conformation", red: "closed conformation") for the disordered, uncrystallized loop comprising of residues 80–89. Residue F83 is indicated in each predicted loop conformation. **b** EC1 crystal structure in the "open conformation" displaying the EC1 residues chosen for mutational screening, ranked and colored according to the number of supporting algorithms. Structure adapted from PDB 6MGA. **c** List of the EC1 residues chosen for mutational screening in (**b**) ranked according to the number of supporting algorithms (interface prediction column). The amino acid substitution(s) are listed for each residue. (EC1, extracellular cadherin domain). Rankings of the residues in the PCDH1 ectodomain for each of the five complimentary algorithms can be found in Supplementary Data 2.

poorly competitive ones (mutations that impair PCDH1 binding; 'poor binders') (Fig. 4c). Within each clade, subgroups that displayed intermediate activity or had mild reduction in binding could also be defined ('intermediate binders'). Similar results were obtained in the infection-inhibition assay (Fig. 5a, b), and a direct comparison of mutation effects in each assay showed that they were highly concordant (Fig. 5c). Altogether, we found 11 EC1 residues whose mutation strongly impairs PCDH1 binding to ANDV Gn/Gc, 10 of which were predicted as interfacial by at least 4/5 algorithms and one (L152) was predicted with lower consent (3/5 algorithms) (Figs. 3b, c and 5c, d). Interestingly, and in accordance with our mapping results, mutation of three residues identified as poor binders in this analysis, reduced recognition by a PCDH1-specific, monoclonal antibody previously shown to block PCDH1-Gn/Gc binding[6], with the strongest effect being observed for D85 (Supplementary Fig. 5). We have thus employed a combination of computational and experimental studies to uncover the ANDV

Gn/Gc-binding surface in PCDH1 EC1. This surface comprises at least 11 residues and is centered around a flexible loop containing a residue (F83) that influences cellular susceptibility to viral entry in a host species-dependent manner.

**Key residues in PCDH1 are required for hantavirus infection**
We identified a PCDH1 EC1 surface patch comprising 11 residues that impair ANDV Gn/Gc binding. Next, we evaluated the roles of two individual residues—F83, which influences cellular susceptibility to viral entry (Fig. 1), and D85, a key residue in the epitope of a mAb that blocks PCDH1 binding (Supplementary Fig. 5). We first generated *PCDH1*-KO U2OS cell lines, stably expressing full-length PCDH1 clones bearing mutations in F83 and D85, or at an adjacent residue not implicated in ANDV Gn/Gc binding, V86. All of the PCDH1 variants resembled WT in their expression level and localization at the cell surface (Supplementary Fig. 6). We evaluated the cells' susceptibility to

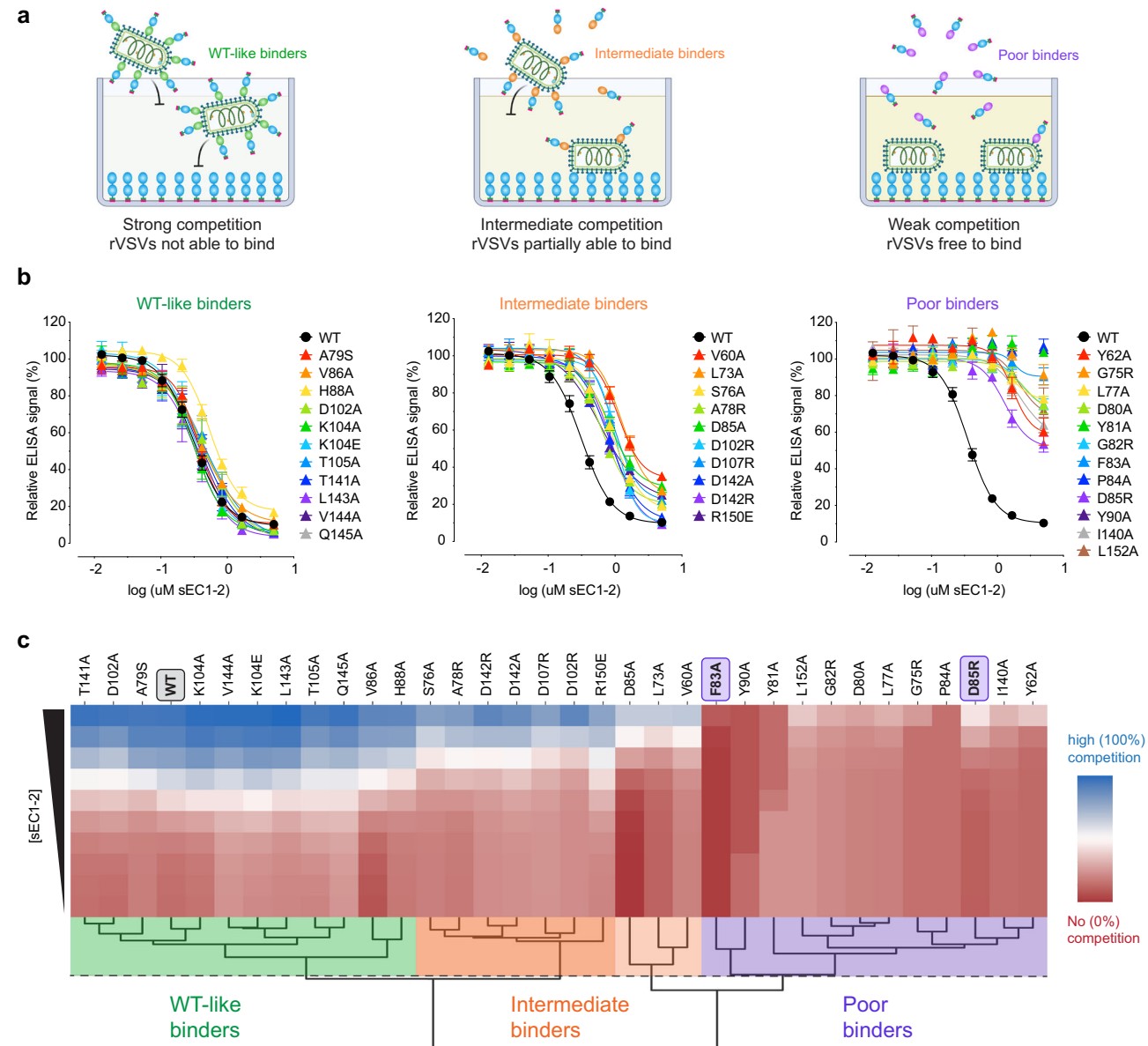

**Fig. 4 | Binding capacity of mutant sEC1-2 to ANDV Gn/Gc. a** Diagram of competition ELISA depicting three different competition outcomes of mutant sEC1-2 proteins' capacity to block rVSV-ANDV-Gn/Gc binding to sEC1-2(WT) coated wells. **b** Competition ELISA using WT and mutant sEC1-2 as competitive reagents to the binding of rVSV-ANDV-Gn/Gc to sEC1-2(WT) coated wells. The ELISA signal was normalized to that obtained without competing sEC1-2. Averages ± SEM: *n* = 6 wells of each dilution examined over three independent experiments, except for: sEC1-2(D102A, K104A, K104E, T105A, V144A, Q145A, D102R, D142A, G75R, Y81A, P84A, I140A) have *n* = 8 wells examined over four independent experiments [sEC1-2(Y81) has *n* = 7 wells for one dilution]. sEC1-2(WT) was used as a reference control for each experiment performed and has *n* = 32 wells of each dilution examined over 16

independent experiments. **c** Hierarchical clustering of WT and mutant sEC1-2 generated from sigmoidal curves of the competition ELISA data in (**a**). The dendrogram shows four clusters, separating the poor binders from the WT-like binders. The dotted line denotes the height at which the dendrogram is cut representing varying degrees of binding strength; WT-like binders (green), intermediate binders (WT-like-to-intermediate binders in dark orange, intermediate-to-poor binders in light orange), and poor binders (purple). The red to blue colorbar ranges from 0 to 100 which is determined by the minimum and maximum values observed in the heatmap. (sEC1-2, soluble extracellular cadherin domains 1 and 2). Figure (**a**) was created with BioRender.com. Source data are provided as a Source Data file.

hantavirus Gn/Gc-dependent infection and found that mutating either F83 or D85 was sufficient to block infection by both rVSV-SNV-Gn/Gc (Fig. 6a) and authentic SNV (Fig. 6b). By contrast, and consistent with the higher affinity of the ANDV Gn/Gc:PCDH1 interaction, a combination of both mutations was necessary to reduce ANDV Gn/Gc-mediated infection (Fig. 6). The V86A PCDH1 variant had little or no effect on entry by either viral glycoprotein (Fig. 6), consistent with its dispensability for PCDH1 recognition by Gn/Gc (Fig. 4b). These findings show that the identified Gn/Gc-binding surface in PCDH1 is required for cell entry by two virulent New World hantaviruses.

**PCDH1 homodimerization is dispensable for Gn/Gc recognition**
Previous work has shown that the PCDH1 ectodomain can form head-to-tail homodimers, driven largely by intermolecular interactions between the EC1 and EC4 domains[17] (Fig. 7a). Although the EC4 and Gn/Gc contact sites in EC1 do not appear to overlap, we considered the possibility that EC1 dimerization may nevertheless impact Gn/Gc recognition. Accordingly, we tested the effects of mutations at EC1 position E137 (Supplementary Fig. 7a), responsible for driving the EC1:EC4 interaction[17], on ANDV Gn/Gc binding. These mutants resembled WT in their capacity to compete for ANDV Gn/Gc binding

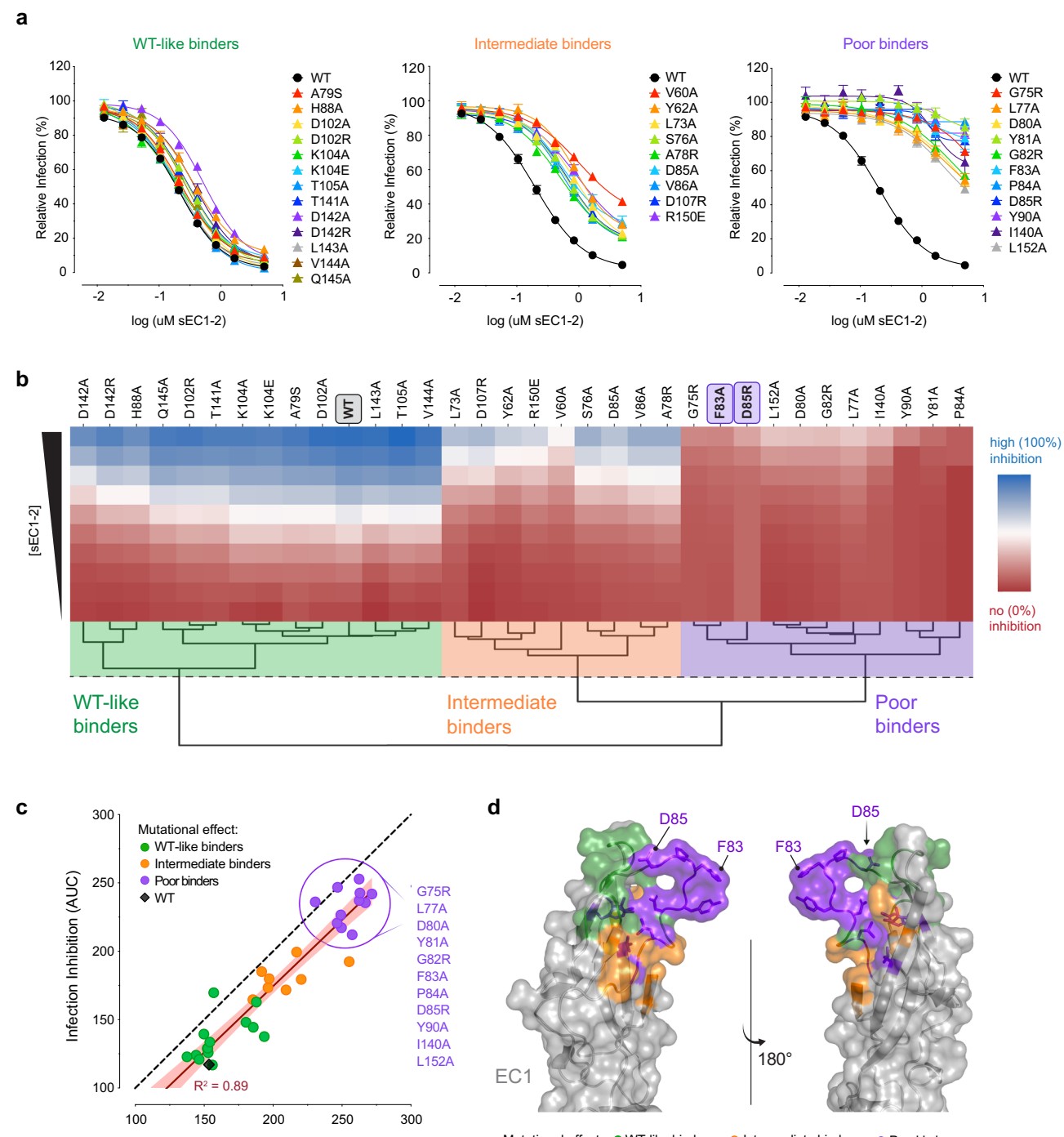

**Fig. 5 | Inhibition of ANDV Gn/Gc-mediated infection by mutant sEC1-2. a** WT and mutant sEC1-2 were tested on their ability to block rVSV-ANDV-Gn/Gc entry in primary human endothelial cells (HUVECs). The infectivity was normalized to that obtained without sEC1-2. Averages ± SEM: *n* = 6 wells of infected cells for each sEC1-2 dilution examined over three independent experiments [sEC1-2(T141A, L143A, V144A, Q145A, D85A, D142A, D142R, Y81A, F83A, D85A, I140A) have *n* = 7, sEC1-2(Y81A) has *n* = 6 for one dilution]. sEC1-2(WT) was used as a reference control and has *n* = 27 wells examined over 13 independent experiments. **b** Hierarchical clustering of WT and mutant sEC1-2 generated from sigmoidal curves of the infection-inhibition assay in (**a**). The dotted line denotes the height at which the dendrogram is cut to obtain three clusters representing varying degrees of inhibition of ANDV Gn/Gc-initiated infection; WT-like inhibition (green), intermediate inhibition (orange), and poor inhibition (purple). The red to blue colorbar ranges from 0 to 100 which is determined by the minimum and maximum values observed in the heatmap. **c** Area under the curve (AUC) for the binding activity of WT and mutant sEC1-2 to rVSV-ANDV-Gn/Gc, as determined by competition ELISA (see Fig. 4b), plotted against AUC values as determined by rVSV-ANDV-Gn/Gc infection-inhibition assay (**a**). The red line denotes the *R* squared value. A list of the sEC1-2 mutants that are classified as poor binders are listed to the right. **d** EC1 crystal structure in the "open conformation" displaying mutated residues representing three degrees of binding strength to ANDV Gn/Gc and inhibition of rVSV-ANDV-Gn/Gc infection. sEC1-2 mutants that bind and inhibit similarly to WT (WT-like binders), green; sEC1-2 mutants that display a mild reduction in binding and inhibition (intermediate binders), orange, and sEC1-2 mutants that display a strong reduction in binding and inhibition (poor binders), purple. Structure adapted from PDB 6MGA. Source data are provided as a Source Data file.

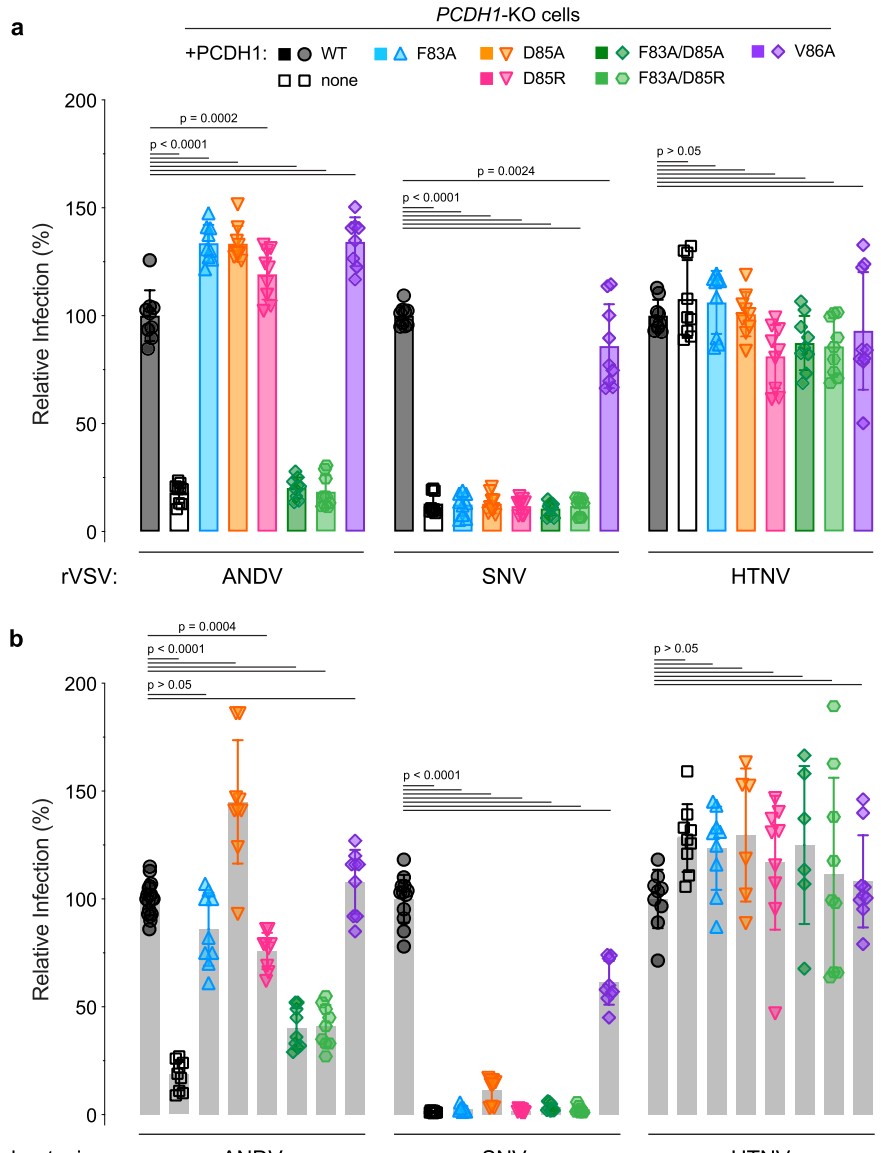

**Fig. 6 | Two key amino acids in PCDH1 mediate entry for SNV and ANDV.**
**a** Relative infectivity of rVSVs bearing ANDV, SNV, or HTNV Gn/Gc on U2OS *PCDH1*-KO cells complemented with WT or mutant PCDH1. The infectivity of each virus was normalized to that obtained in U2OS *PCDH1*-KO cells complemented with WT PCDH1. Means ± SD: *n* = 9 wells of infected cells examined over three independent experiments (rVSV-ANDV-Gn/Gc infection on U2OS *PCDH1*-KO cells complemented with V86A had *n* = 8). Infectivities were compared by one-way ANOVA with Dunnett's test for multiple comparisons. **b** Relative infectivity of authentic ANDV, SNV, or HTNV on the cell lines described in (**a**). The infectivity of each virus was normalized to that obtained in U2OS *PCDH1*-KO cells complemented with WT PCDH1. Means ± SD: *n* = 9 infected wells were examined over three independent experiments (HTNV infection on U2OS *PCDH1*-KO cells complemented with D85A and F83A/D85A had *n* = 6 wells of infected cells examined over two independent experiments). For ANDV and SNV infection on the control cell line, U2OS *PCDH1*-KO cells complemented with WT, *n* = 12 (SNV) and *n* = 18 (ANDV) wells of infected cells were examined over four independent experiments. Infectivities were compared by one-way ANOVA with Dunnett's test for multiple comparisons. Source data are provided as a Source Data file.

(Fig. 7b). To further examine the effect of PCDH1 dimerization on its hantavirus receptor activity, we stably expressed PCDH1 lacking the EC4 domain in *PCDH1*-KO U2OS cells (Supplementary Fig. 7b) and tested their susceptibility to Gn/Gc-mediated infection (Fig. 7c). This "monomer-only" PCDH1(ΔEC4) supported ANDV and SNV Gn/Gc-dependent entry in a manner similar to WT PCDH1.

PCDH1 interaction and homodimerization in trans is proposed to regulate cell adhesion between neighboring cells, suggesting that both monomers and dimers co-exist at the cell surface[17]. Although our preceding results indicated that hantaviruses could efficiently recognize and use PCDH1 monomers, they did not exclude the possibility that PCDH1 dimers may also provide suitable entry receptors. To test this, we engineered sEC1-4 variants that are intrinsically either

monomers or dimers (Fig. 7d). Specifically, we generated obligate monomers by introducing the K455E mutation in EC4 to abrogate its salt bridge with E137 in EC1, disrupting the EC1:EC4 interaction[17]. To generate obligate dimers, we engineered a sEC1-4 variant in which structurally apposed residues, T141 and G337 in EC1 and EC4, respectively, were mutated to C to afford intersubunit disulfide formation. sEC1-4(T141C/G337C) displayed shifts in electrophoretic mobility relative to sEC1-4 (WT) and sEC1-4(K455E) monomer at nonreducing conditions in denaturing (Fig. 7e) and native polyacrylamide gels (Fig. 7f), concordant with its formation of a disulfide-bonded dimer. Pre-titrated amounts of each sEC1-4 protein bound ANDV Gn/Gc; however, dimeric sEC1-4(T141C/G337C) appeared to recognize ANDV Gn/Gc less efficiently than its WT and obligate-monomer counterparts

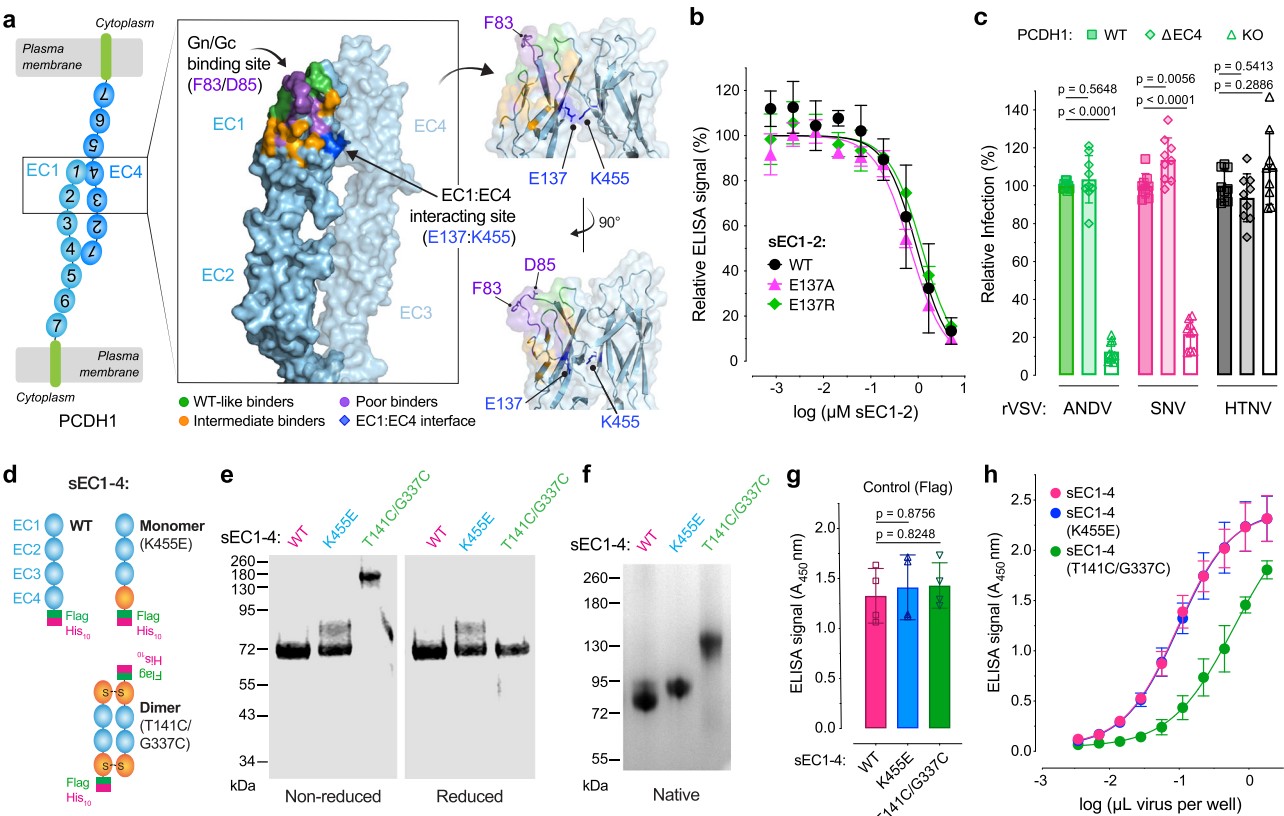

**Fig. 7 | Monomeric and dimeric PCDH1 provide suitable entry receptors for ANDV. a** Crystal structure of the proposed anti-parallel EC1-4 trans-dimer. Structure is in the "open conformation" displaying residues representing three degrees of binding strength to ANDV Gn/Gc. Residues that when mutated bind similarly to WT (green); those that display a mild reduction in binding (orange); and those that display a strong reduction in binding (purple). The Gn/Gc binding site relative to the EC1:EC4 adhesive interface (dark blue) is indicated. An alternative view of the EC1:EC4 binding interface is shown to the right. Structure adapted from PDB 6MGA. **b** Competition ELISA using WT and mutant sEC1-2 as competitive reagents to the binding of rVSV-ANDV-Gn/Gc to WT sEC1-2 coated wells. Averages ± SD: $n = 4$ wells of each sEC1-2 dilution examined over two independent experiments [$n = 3$ for one of the dilutions of sEC1-2(E137R)]. **c** Relative infectivity of rVSVs bearing ANDV, SNV, or HTNV Gn/Gc on U2OS *PCDH1*-KO cells complemented with WT or ΔEC4 PCDH1. The infectivity of each virus was normalized to that obtained in U2OS *PCDH1*-KO cells complemented with WT PCDH1. Means ± SD: $n = 9$ wells of infected cells

examined over three experiments. **d** Schematic representation of WT or mutant sEC1-4 proteins forming monomers or dimers. **e** Non-reduced and reduced purified WT and mutant sEC1-4 were separated on an SDS-polyacrylamide gel and visualized by Coomassie Brilliant Blue staining. kDa, kilodalton. **f** Non-reduced samples in (**e**) were run on a native-polyacrylamide gel and visualized as in (**e**). A representative gel from one experiment of two independent experiments is shown for (**e**) and (**f**). **g** ELISA detecting WT and mutant sEC1-4 coated plates, using an anti-Flag-HRP antibody. Mean ± SD: $n = 4$ wells examined over two independent experiments. (**h**) Capacity of rVSV-ANDV-Gn/Gc to bind to WT or mutant sEC1-4 coated plates. Done in parallel with (**g**). Mean ± SD: $n = 4$ wells of each viral particle dilution examined over two independent experiments. Infectivities (**c**) and ELISA signal (**g**) were compared by one-way ANOVA with Dunnett's test for multiple comparisons. (sEC1-4, soluble extracellular cadherin domains 1–4). Source data are provided as a Source Data file.

(Fig. 7g, h). These findings suggest that both monomeric and dimeric forms of PCDH1 provide suitable entry receptors for hantaviruses but also raise the possibility that viral particles may preferentially engage PCDH1 monomers.

## PCDH1 mutations protect hamsters against ANDV challenge

We previously demonstrated that ANDV infection and virulence was highly attenuated in Syrian hamsters engineered to lack PCDH1 expression[6]. Although these studies identified PCDH1 as a critical requirement in ANDV multiplication and pathogenesis per se, they did not specifically address if this requirement arose from PCDH1's role as an entry receptor that directly engages the viral glycoprotein complex. To test the latter hypothesis, we used CRISPR/Cas9 genome engineering to introduce a F83A/D85R double mutation at the *PCDH1* locus in Syrian hamsters (Fig. 8a). Biallelic *PCDH1(F83A/D85R)* animals expressed levels of PCDH1 in the lung similar to their WT counterparts (Fig. 8b).

Interestingly, we also identified founder animals with an allele encoding a larger 10 amino acid deletion in the Gn/Gc-binding surface: *PCDH1(S76G, ΔL77–D85)* (Fig. 8a). This fortuitous mutant [hereafter,

PCDH1(10a.a.)] abrogated PCDH1's receptor function in vitro, but at the expense of a partial reduction in its steady state expression level, which was also observed in vivo (hamster lung tissue) (Fig. 8b and Supplementary Fig. 8a, c). Nevertheless, PCDH1(10a.a.)-expressing cell subpopulations, sorted to approximate WT PCDH1 expression levels, remained resistant to viral entry (Supplementary Figs. 8b–c, 9) suggesting that the loss of 7 of 11 residues implicated in ANDV and SNV Gn/Gc recognition, and not reduced PCDH1 expression, accounts for PCDH1(10a.a.)'s loss of receptor activity. Considering that *PCDH1(10a.a.)* afforded a genotype intermediate to that of the *PCDH1(F83A/D85R)* and *PCDH1(KO)* alleles, we evaluated both CRISPR/Cas9 knock-in Syrian hamsters in ANDV challenge.

We challenged WT and knock-in hamsters intranasally with a lethal dose of ANDV and monitored their survival for up to 35 days post-exposure (Fig. 8c and Supplementary Fig. 10). Hamsters carrying modified *PCDH1* alleles (either *PCDH1(F83A/D85R)* or *PCDH1(10a.a.)*) were largely protected against a lethal outcome of ANDV challenge while the control animals carrying WT *PCDH1* succumbed to infection. Histochemical analysis of lung tissues at 15 days post ANDV exposure revealed reduced inflammation and expansion

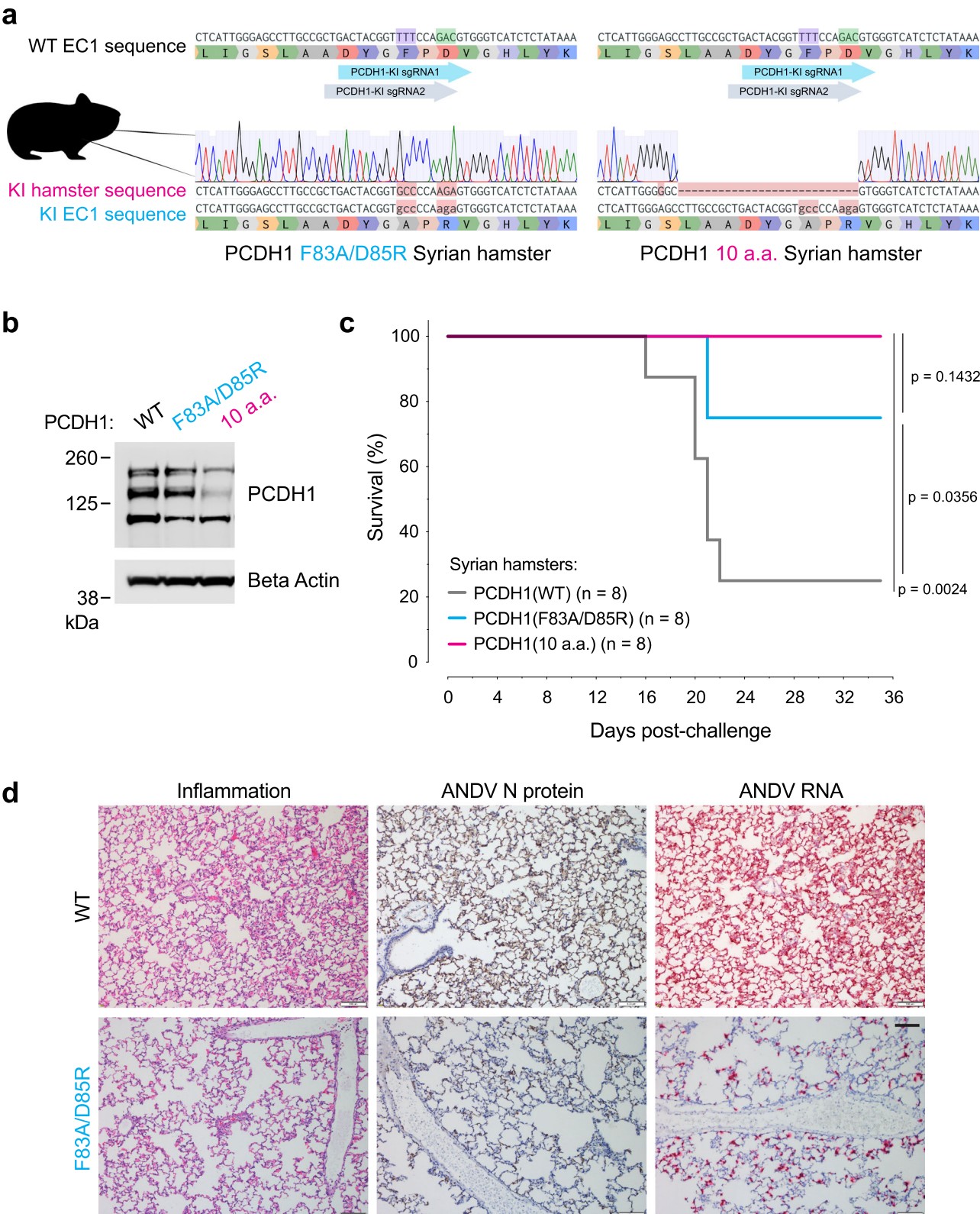

of alveolar septa due to monocyte infiltration (left panels), lower viral RNA (red staining, right panels) and nucleoprotein levels (tan staining, middle panels) in hamsters expressing PCDH1(F83A/D85R) compared to WT hamsters (Fig. 8d). Seroconversion, indicative of an ANDV-specific IgG response, was observed in all surviving hamsters, indicating they had all been exposed to the virus (Supplementary Fig. 10). Our Syrian hamster challenge study strongly supports our hypothesis that a direct Gn/Gc:PCDH1 engagement is a critical

requirement for ANDV multiplication and lethal, in vivo HCPS-like pathogenesis.

## Discussion
Virus–receptor interactions can influence viral entry, cell and tissue tropism, viral host range and pathogenesis, and afford attractive targets for antiviral countermeasures[23–32]. We previously demonstrated that PCDH1 is an essential entry host factor for New World

**Fig. 8 | Two point mutations in PCDH1 confer protection of Syrian hamsters against a lethal ANDV challenge. a** Reference nucleotide and amino acid sequence of PCDH1-EC1 Syrian hamster (WT, above) and representative sequences and trace files of Syrian hamsters after CRISPR-Cas9 genome editing [PCDH1(F83A/D85R), lower left] and [PCDH1(10a.a.) lower right]. The nucleotides encoding the corresponding human PCDH1-EC1 Gn/Gc-interacting residues, are highlighted: F83 in purple and D85 in green along with the location of the single guide RNAs (KI, knock-in; sgRNA, single guide RNA). **b** Immunoblot detecting PCDH1 in lung tissue lysates from WT or CRISPR knock-in mutant Syrian hamsters. Antibody targets PCDH1's cytoplasmic tail. kDa, kilodalton. A representative blot from a single experiment of two independent experiments is shown. Uncropped blots in Source Data. **c** Syrian hamster ANDV challenge. Groups of WT, PCDH1(F83A/D85R), and

PCDH1(10a.a.) CRISPR knock-in mutant hamsters were inoculated intranasally with ANDV (2,000 PFU). Mortality was monitored and hamsters were euthanized on day 35 post-exposure. One experiment was performed, with $n = 8$ hamsters for each group. Data was analyzed using two-sided, log-rank Mantel–Cox test. **d** Lung sections from WT and PCDH1(F83A/D85R) hamsters were collected 15 days post ANDV exposure. Representative histochemical images indicate inflammation in pulmonary tissue (left), ANDV nucleoprotein (N) (middle, tan staining), and ANDV RNA (right, red staining, detected by in situ hybridization). Representative images from one experiment from one out of three hamsters from each group are shown. Scale bars represent 100 μm. Figure (**a**) includes an image from Flaticon.com. Source data are provided as a Source Data file.

---

hantaviruses[6,7]. Herein, we identified residues in the EC1 domain of PCDH1 critical for ANDV and SNV Gn/Gc engagement, including one that influences SNV cellular host range and demonstrated that Gn/Gc:PCDH1 recognition, not just PCDH1 per se, is required for the development of lethal ANDV infection in Syrian hamsters. We conclude that PCDH1 is a bona fide entry receptor for ANDV and SNV, the primary etiologic agents of HCPS in the Americas, and is likely to play such a role for other New World hantaviruses shown to utilize PCDH1 for entry[6,7].

We leveraged the recently published crystal structure of PCDH1 domains 1-4 (EC1-4)[17] and structure-based prediction of interfacial residues to generate a list of candidate virus-contacting residues in PCDH1 EC1. Quantitative assessment of this panel of site-directed EC1 mutants, through receptor-binding and -blocking assays, showed that a membrane-distal EC1 surface centered around a flexible loop makes key contacts with Gn/Gc during hantavirus–receptor recognition (Figs. 4, 5). The identified key residues might either directly contact Gn/Gc or might indirectly contribute to Gn/Gc binding through intra-molecular interactions that maintain the local geometry of EC1, as is likely the case for more buried residues (e.g., L152). Our findings also shed light on differences in the mechanisms of PCDH1 recognition by ANDV and SNV Gn/Gc. While ANDV Gn/Gc binds with higher avidity to PCDH1 than its SNV counterpart (Fig. 2b)[6], both Gn/Gc proteins displayed similar binding patterns toward our PCDH1 mutant panel (Fig. 6), suggesting that they largely share their key PCDH1 contacts. Instead, the differences in binding affinity between ANDV and SNV Gn/Gc (and by extension, the glycoproteins from other New World hantaviruses) likely arise from sequence variations in Gn/Gc's yet-unmapped PCDH1-binding site (also see below). We further speculate that more divergent residues at those or adjacent positions with Gn/Gc account for the failure of Old World hantavirus Gn/Gc proteins to engage with PCDH1. Whether non-PCDH1–using Old World hantaviruses recognize their putative receptors through the same or distinct surfaces on Gn/Gc remains to be determined.

Herein, we provide evidence that hantavirus Gn/Gc:PCDH1 recognition can impact cellular host range. Specifically, we observed that murine endothelial cells are refractory to SNV Gn/Gc-dependent entry (Fig. 1a) and mapped this human-murine difference in susceptibility to a sequence variation at a single residue in EC1, residue 83, that modulates Gn/Gc-PCDH1 engagement (Fig. 1f). This PCDH1 ortholog-dependent effect on entry was not observed for ANDV, despite a discernible effect on receptor binding (Figs. 1f, 2f), presumably because ANDV Gn/Gc, unlike its SNV counterpart, retains sufficient binding avidity for murine PCDH1 on cell surfaces. Residue F83 afforded us a nidus to more comprehensively map the Gn/Gc-interacting surface in EC1 and identify 10 additional surface-exposed residues that are key for virus–receptor recognition. Indeed, mutation of F83 in combination with the adjacent D85 could further ablate ANDV Gn/Gc-mediated entry and infection in vitro (Fig. 6) and significantly reduced mortality from lethal ANDV challenge in Syrian hamsters (Fig. 8c). Hamsters carrying a larger 10-amino acid disruption in PCDH1, encompassing F83, D85, and additional interfacial residues that individually and

collectively impacted Gn/Gc-PCDH1 binding, phenocopied PCDH1-KO hamsters in being fully resistant to lethal virus challenge (Fig. 8c), likely by further reducing receptor recognition and cellular susceptibility to viral infection in vivo, as observed in cell culture (Supplementary Fig. 8c).

Our findings point to a role for New World hantavirus Gn/Gc:PCDH1 recognition in influencing viral host range in nature and suggest at least three new avenues for exploration, including: (i) sequencing and functional analyses of Gn/Gc (from circulating viruses) as well as PCDH1 orthologs in geographically/ecologically distinct populations of established hantavirus hosts (e.g., the pygmy rice rat for ANDV); (ii) investigations of virus–receptor compatibility in natural hantavirus infections of rodents not traditionally considered to be viral reservoirs (e.g., SNV in wild-caught house mice[33]); and (iii) experimental cross-species infections to uncover novel virus–receptor mismatches that may pinpoint additional determinants of viral host range[34]. Further, the virus–receptor interactions defined herein may illuminate our understanding of hantavirus infection and pathogenesis in the available animal models[35–41] and facilitate the development of new models to study virus–host interactions and test countermeasures. As a case in point, the New World hantaviruses ANDV and SNV do not appear to infect laboratory mouse strains, possibly due in part to the incompatibility between Gn/Gc and murine PCDH1 (especially for SNV) that we uncovered in this study. We speculate that transgenic mice bearing compatible PCDH1(L83F) alleles may sustain viral replication by removing a key host barrier to viral entry.

Disruptions in PCDH1's cellular functions have been linked to the dysfunction of the airway epithelial barrier in respiratory diseases such as asthma[10]. This raises the tantalizing possibility that the cellular functions of PCDH1 are intertwined with the pathogenesis of severe pulmonary disease caused by ANDV and SNV infections in humans. Although PCDH1's endogenous roles and their molecular mechanisms remain poorly understood, its capacity to form trans-dimers through EC1:EC4 domain interactions is proposed to be central to its adhesive capacity[17]. Here, we found that hantaviruses recognize a surface in PCDH1 EC1 that is distinct from the EC1-EC4 adhesive interface (Fig. 7a, b), suggesting that virus–receptor interaction during entry, or putative glycoprotein–receptor interactions in infected cells, are unlikely to interfere directly with PCDH1 dimerization or vice versa. Studies with recombinant PCDH1 mutants engineered to form only monomers or only dimers supported this hypothesis (Fig. 7h), as did the observation that deletion of EC4 (and abrogation of EC1-EC4 association) did not impair viral entry (Fig. 7c). However, we did also obtain preliminary evidence that PCDH1 dimerization (or attendant conformational changes) may subtly disfavor Gn/Gc interaction (Fig. 7h), raising the possibility that Gn/Gc expression in infected cells may shift the PCDH1 monomer-dimer equilibrium by preferentially sequestering monomers at the cell surface. More work is needed to test this idea as well as simpler (e.g., PCDH1 down-regulation) and more complex scenarios (e.g., Gn/Gc-induced changes in PCDH1 conformation or transmembrane signaling), through which hantavirus infection could perturb PCDH1's endogenous functions in the mammalian airway.

## Methods

### Cells

Human pulmonary microvascular endothelial cells (HPMEC, Promocell, catalog No. C-12281) and mouse lung microvascular endothelial cells (MLMEC, Cell Biologics, catalog No. BALB-5012) were cultured in MV2 Endothelial Cell Growth medium (Promocell) and Complete Mouse Endothelial Cell Medium (Cell Biologics), respectively. Human osteosarcoma U2OS cells (ATCC, catalog No. HTB-96) and *PCDH1*-knockout (KO) U2OS cells, generated as described in Jangra et al.[6], were cultured in modified McCoy's 5A media (Thermo Fisher), supplemented with 10% fetal bovine serum (FBS, Atlanta Biologicals), 1% GlutaMAX (Thermo Fisher), and 1% penicillin-streptomycin (Pen-Strep, Thermo Fisher). Human umbilical vein endothelial cells (HUVEC, Lonza, catalog No. C2519A) were cultured in EGM media supplemented with EGM-SingleQuots (Lonza). Embryonic kidney fibroblast 293T cells (ATCC, catalog No. CRL-3216), grivet kidney Vero cells (ATCC, catalog No. CCL-81), and grivet kidney Vero E6 cells (ATCC, catalog No. CRL-1586) were cultured in high-glucose Dulbecco's modified Eagle medium (DMEM), supplemented with either 10% (293 T cells) or 2% (Vero and Vero E6 cells) FBS (Atlanta Biologicals), 1% GlutaMAX (Thermo Fisher), and 1% Pen-Strep (Thermo Fisher). All adherent cell lines were maintained in a humidified 37 °C, 5% $CO_2$ incubator. Freestyle™–293-F suspension cells (Thermo Fisher, catalog No. R79007) were maintained in FreeStyle™ 293 expression medium (Thermo Fisher) using shaker flasks at 115 rpm, 37 °C, and 8% $CO_2$. Drosophila Schneider 2 cells (Thermo Fisher/Gibco, catalog No. R69007), stably expressing soluble Andes virus (ANDV) Gn$^H$/Gc[22], were maintained in serum-free insect cell medium (GE Healthcare HyClone) using spinner flasks at 28 °C.

### rVSVs and infections

Replication-competent, recombinant vesicular stomatitis Indiana viruses (rVSVs) expressing an eGFP reporter and bearing VSV G or hantavirus Gn/Gc of either ANDV (NP_604472.1), Sin Nombre virus (SNV) (NP_941974.1), or Hantaan virus (HTNV) (NP_941978.1) with the modifications described in Slough et al.[42], were generated using a plasmid-based rescue system in 293T cells and propagated on Vero cells as described previously[43,44]. The sequences of all rVSV glycoproteins were amplified from viral genomic RNA by RT-PCR and analyzed using Sanger sequencing. For infection experiments; HPMECs and MLMECs were exposed to virus at a multiplicity of infection (MOI) of 0.3 infectious units (IUs) per cell for 1 h prior to stopping infection with NH$_4$Cl (20 mM). U2OS *PCDH1*-KO cells, complemented with wild-type (WT) or mutant PCDH1, were exposed to virus at an MOI of 0.03 IUs per cell and allowed infection to proceed for 12–14 h. For all infections, cells were fixed 12–14 h post-infection with 4% formaldehyde (Sigma) and counter-stained with Hoechst nuclear stain (Invitrogen). Number of infected cells (GFP+ cell counts) and viral infectivity (%GFP+ cells) was measured by automated enumeration of eGFP-expressing cells from captured images from multiwell plates, using a CellInsight CX5 automated fluorescence microscope with onboard HCS Studio software (Thermo Fisher, V.6.6.0) or Cytation5 cell imaging multi-mode reader with Agilent Biotek Gen5 Microplate Reader and Imager software (BioTek, V.3.2). For sEC1-2 infection-inhibition experiments, pre-titrated amounts of rVSV particles (MOI of 0.1) were incubated with increasing concentrations of WT or mutant sEC1-2 at room temperature (RT) for 1 h prior to the addition to HUVEC monolayers in 96-well plates. 12–14 h post-infection, cells were fixed, stained, and the number of infected cells was measured as described above.

### Authentic hantaviruses and infections

ANDV strain Chile-9717869, SNV strain CC107, and HTNV strain 76-118 were propagated in Vero E6 cells as described previously[35,45]. Hantavirus infections were performed, and infected cells were immunostained for viral antigen, as described previously[43]. Briefly, U2OS *PCDH1*-

KO cells, complemented with WT or mutant PCDH1, were exposed to virus at an MOI of 0.5 (ANDV), 1.5 (SNV), or 3 (HTNV) plaque-forming units (PFU) per cell, and viral infectivity was determined by immunostaining of formalin-fixed cells at 72 h post-infection using a 1:5,000 dilution of rabbit polyclonal antibodies (pAbs) detecting ANDV (NR-9673, BEI Resources), HTNV (NR-12152, BEI Resources) or SNV (NR-9674, BEI Resources) nucleoproteins. Images were acquired at 20 fields per well, with a 20x objective on an Operetta high-content imaging device (PerkinElmer). Images were analyzed with a customized scheme built from image analysis functions present in Harmony software (V.4.8) and the percentage of infected cells was determined using the analysis functions.

### PCDH1 EC1 sequence alignment

Alignment of amino acid sequences of the EC1 domain of PCDH1 from six different species was generated by Clustal Omega. The PCDH1 sequences used for the alignment, along with their GenBank accession numbers, were as follows: *Mesocricetus auratus* (golden or Syrian hamster), XP_021082321.1; *Peromyscus maniculatus bairdii* (prairie deer mouse), XP_015863073.1; *Mus musculus* (house mouse), NP_001390724.1; *Rattus norvegicus* (Norway rat), XP_038953281.1; *Macaca fascicularis* (crab-eating macaque), XP_045250283.1; *Homo sapiens* (human), NP_002578.2.

### PCDH1 EC1 loop modeling

The EC1 missing loop (residues 80-89) in the crystal structure of human PCDH1 was modeled using MODELLER (V.9.22) through the Chimera (V.1.14) plugin[17,46,47]. All 10 modeled loop conformations scored similarly, based on their GA341 and zDOPE scores[48,49]. The two most divergent loop conformations were selected as representative loop conformations to perform structure-based interfacial residue predictions on.

### Interfacial residue predictions in the PCDH1 ectodomain

Five complementary structure-based tools [(PredUs (V.2.0), SPPIDER (V.2), consPPI (V.1.0), PINUP (V.1.0), and ProMate (V.2)][50–54] were used to predict interfacial residues in the PCDH1 ectodomain crystal structure, with two modeled alternative conformations for the missing loop (residues 80-89; see above). For any given prediction method, the results for each of the two versions of the PCDH1 ectodomain structure (Pred$_A$, Pred$_B$) were combined into a single set of predictions (Pred$_A$ ∪ Pred$_B$). Predicted interfacial residues were then ranked based on the number of supporting algorithms (range 0–5).

### Cloning soluble PCDH1 variants

Constructs encoding soluble (secreted) PCDH1 variants (sEC1-2 and sEC1-4) were generated by cloning the following sequences into the pcDNA3.1 mammalian expression vector (Thermo Fisher): EC1-EC2 (residues 1–284)[6] or EC1-EC4 (residues 1–503), each in frame with a C-terminal GSG linker, followed by Myc, Flag, and deca-histidine tags. Each construct also retained the endogenous PCDH1 N-terminal signal sequence (residues 1–57). PCDH1 point mutations were cloned into the pcDNA3.1 plasmid using standard molecular biology techniques. To avoid free cysteine-mediated, non-specific protein–protein interactions, sEC1-4 variants included a C432S mutation. The C432S mutation did not have an observable effect on the production of the soluble protein or on the binding to rVSV-ANDV-Gn/Gc. The sequences of all the plasmid inserts were confirmed by Sanger sequencing.

### Expression and purification of soluble PCDH1 variants

Soluble PCDH1 variants cloned into pcDNA3.1 (see above) were expressed in 293 F cells in shaker flasks by transient transfection with linear polyethyleneimine (Polysciences) and purified by nickel-chelation chromatography. Cell cultures were incubated at 37 °C and 8% $CO_2$ for six days post-transfection. Cell supernatants were clarified

and stirred overnight at 4 °C with proteinase inhibitor (Sigma-Aldrich) and nickel-NTA resin (Qiagen) at 0.3 mL packed resin per 50 mL cell supernatant. Nickel-NTA beads were then collected, washed with phosphate buffer saline (PBS) containing 50 mM imidazole, and eluted with PBS containing 250 mM imidazole. The eluted protein was buffer-exchanged with PBS, concentrated, and stored in aliquots at −80 °C. The purity of the secreted PCDH1 variants was determined by size-exclusion chromatography (SEC) and/or either SDS-PAGE or Native-PAGE gels, stained with Bio-Safe™ Coomassie G-250 Stain (Bio-Rad) and imaged on a LI-COR Odyssey® Fc Imager (LI-COR Biosciences, LICOR Image Studio software, V.1.0.19). For analytical SEC, a Superdex S200 (10/300) column was equilibrated in PBS and calibrated with Gel Filtration Standard (Bio-Rad) composed of thyroglobulin (MW 670 kDa), bovine γ-globulin (MW 158 kDa), chicken ovalbumin (MW 44 kDa), horse myoglobin (MW 17 kDa), and vitamin B12 (MW 1.35 kDa).

## Expression and purification of soluble ANDV Gn^H/Gc

The generation of soluble ANDV Gn^H/Gc (sGn^H/Gc) has previously been described in Serris et al.[22]. Briefly, we used a plasmid coding for ANDV Gn^H and the ectodomain of Gc joined by a flexible linker. To facilitate purification, a double Strep-tag was included in the C-terminus of Gc. This plasmid was used to generate a stable line of S2 insect cells. These cells were grown on 1 L spinners, and sGn^H/Gc expression was induced using 4 μM CdCl₂ for five days. sGn^H/Gc was purified from the supernatant using a combination of affinity and size exclusion chromatography. The purity of the purified protein was determined by SDS-PAGE gel, stained with Bio-Safe™ Coomassie G-250 Stain (Bio-Rad).

## rVSV:PCDH1 competition ELISA

The capacity of sEC1-2 mutants to compete with the binding of sEC1-2(WT) to rVSVs, bearing ANDV or SNV Gn/Gc, was determined by a competition capture ELISA. High-protein binding 96-well ELISA plates (Corning) were coated with purified sEC1-2 (100 ng/well) overnight at 4 °C, washed briefly with PBS, and blocked with 5% nonfat dry milk in PBS (1 h at RT). Pre-titrated amounts of rVSV-ANDV-Gn/Gc and rVSV-SNV-Gn/Gc, were membrane-labeled with a short-chain phospholipid probe, functional-component spacer diacyl lipid conjugated to biotin (FSL-biotin (5 μg/mL); Sigma-Aldrich) for 1 h at 37 °C. The rVSVs were pre-incubated with serial 2x- or 3x-diluted sEC1-2(WT or variants) for 1 h at RT prior to their incubation with sEC1-2 coated wells (1 h at 37 °C). Bound rVSVs were detected by incubation with Pierce™ High Sensitivity Streptavidin-horseradish peroxidase (HRP) conjugate (1:10,000 dilution, Thermo Scientific). ELISA signal was developed using 1-Step™ Ultra TMB-ELISA substrate solution (Thermo Scientific) and measured at an absorbance at 450 nm on a Perkin Elmer Wallac 1420 Victor2™ microplate reader or Cytation5 cell imaging multi-mode reader (Agilent BioTek Gen5 Microplate Reader and Imager software, V.3.2).

## Gn/Gc:PCDH1 binding ELISA

The capacity of soluble ANDV Gn^H/Gc (sGn^H/Gc) or rVSVs bearing ANDV, SNV, or HTNV Gn/Gc to recognize sEC1-2, and rVSVs bearing ANDV Gn/Gc to recognize sEC1-4 was determined by capture ELISA. High-protein binding 96-well ELISA plates (Corning) were coated with purified sEC1-2 or sEC1-4 (100 ng/well) overnight at 4 °C, washed briefly with PBS, and blocked with 5% nonfat dry milk in PBS (1 h at RT). ANDV sGn^H/Gc was serial diluted twofold with an initial starting dilution of 6.25 μg/well. Equivalent amounts of rVSV-ANDV-Gn/Gc, rVSV-SNV-Gn/Gc, and rVSV-HTNV-Gn/Gc were membrane-labeled with FSL-biotin as described above, and serially diluted (twofold). ANDV sGn^H/Gc and rVSVs were added to either sEC1-2 or sEC1-4 coated plates and incubated for 1 h at 37 °C. Bound ANDV sGn^H/Gc was detected by incubation with Strep-Tactin®-HRP conjugate (1:5,000 dilution, IBA Lifesciences) and rVSVs were detected by incubation with Pierce™ High Sensitivity Streptavidin-HRP conjugate (1:10,000 dilution, Thermo Fisher). Coated sEC1-4 protein was detected by incubation

with an anti-Flag, clone M2 monoclonal antibody (mAb)-HRP conjugate (1:1,000 dilution, catalog No. A8592, Sigma-Aldrich) (1 h at 37 °C). ELISA signal was developed and measured as noted above.

## Monoclonal antibody:PCDH1 binding ELISA

To determine the capacity of an infection-inhibiting, EC1-specific mAb to bind to sEC1-2 variants, ELISA plates were coated with serial twofold dilutions of WT or mutant sEC1-2 (starting concentration at 400 ng/well) overnight at 4 °C. After briefly washing with PBS, wells were blocked with 5% nonfat dry milk in PBS (1 h at RT) and incubated with anti-EC1 mAb-3305 (1 μg/mL, Donnelly Centre and Department of Molecular Genetics, University of Toronto) for 1 h at RT. After washing with PBS, the bound antibody was detected by incubation with an anti-human IgG HRP pAb (1:10,000 dilution, catalog No. AP112, Sigma-Aldrich) for 1 h at RT. ELISA signal was developed using 1-Step™ Ultra TMB-ELISA substrate solution (Thermo Scientific) and measured at an absorbance of 450 nm on a Cytation5 cell imaging multi-mode reader (Agilent BioTek Gen5 Microplate Reader and Imager software, V.3.2).

## Hierarchical data clustering

An in-house algorithm was used to perform hierarchical clustering of two experimental readouts: sEC1-2 competition ELISA (absorbance at 450 nm) and infection-inhibition assay (GFP-positive cells [%]). A sigmoidal function (A) (see below) was fitted to the normalized experimental readouts using a non-linear least square analysis as implemented in the SciPy package[55]. Hierarchical clustering and a heatmap comparing each sigmoidal curve to all other sigmoidal curves were built using the clustermap (method = average; metric = Euclidean distance) function as implemented in the seaborn python package (https://seaborn.pydata.org/index.html):

$$y = y_{min} + \left[ (y_{max} - y_{min})/(1 + 10^{\log 10(EC50-x)xHill}) \right]$$

where $y$ corresponds to the experimental readout (ER) (relative ELISA signal or percent infectivity); $y_{min}$ and $y_{max}$ are the minimum and maximum ERs, respectively; $EC_{50}$ is the value that gives half-maximum ER (half-$y_{max}$); Hill describes the slope of the curve, and $x$ is the amount of soluble protein, $\log_{10}$(sEC1-2), at the particular $y$ (ER). (Original code can be found at: https://github.com/chandranlab/pcdh1_interface.git or in the Figshare database under accession code https://doi.org/10.6084/m9.figshare.23469236[56].

## Stable cells expressing PCDH1 variants

cDNA constructs encoding full-length human PCDH1 (isoform 1, Genbank accession number NM_002587) or lacking the fourth extracellular cadherin (EC) repeat ΔEC4 (residues 328-446) were synthesized in frame with Myc and Flag epitope tags at the C-terminus (Epoch Biolabs or Twist Bioscience) and cloned into the pBABE-puro retroviral vector[57]. Residue numbers are based on the full-length sequence of PCDH1 starting from the signal sequence (i.e., residue D85 in this study matches PDB 6MGA residue D28, and E137 in this study matches PDB 6MGA residue E80). Mutations for the PCDH1 variants were introduced into the pBABE-puro-PCDH1 plasmid using standard molecular techniques and confirmed by Sanger sequencing. Human U2OS *PCDH1*-KO cells ectopically expressing the above PCDH1 variants were generated by transduction with pBABE-puro-based retroviral vectors. Retroviruses packaging the transgenes were produced by transfecting 293T cells[43], and target cells were directly exposed to sterile-filtered, retrovirus-laden supernatants in the presence of polybrene (6 μg/mL). Transduced U2OS cell populations were selected with puromycin (2 μg/mL), and transgene expression was confirmed by immunostaining. MLMECs were also transduced as above but were not subjected to antibiotic selection.

## Detection of PCDH1 surface expression by flow cytometry

Human U2OS cells expressing variant PCDH1 were seeded in 6-well plates 24 h prior to immunostaining. Cells were chilled on ice for 10 min (min) and blocked with chilled PBS/10% FBS for 30 min at 4 °C. Surface PCDH1 was stained using human anti-EC7 mAb-3677 (5 µg/mL, Donnelly Centre and Department of Molecular Genetics, University of Toronto) followed by anti-human IgG Alexa Fluor™ 488 pAb (1:500 dilution, catalog No. A-11013, Thermo Fisher) for 1 h at 4 °C. After washing, cells were resuspended in PBS/2% FBS, and those intended to be sorted were stained with TO-PRO™3 Ready Flow™ Reagent (Invitrogen) to identify and exclude dead cells. Cells were passed through a 0.41 µm Nylon Net Filter (Millipore) and analyzed using an LSRII Flow Cytometer (BD Biosciences, CellQuest Pro software, V.6.1) and FloJo V.10 software. Subpopulations of cells expressing PCDH1(10a.a.) were isolated by FACS (NanoCellect WOLF Cell Sorter, WOLFViewer software, V.2.4) and verified following the staining method described above.

## Immunofluorescence microscopy

For PCDH1 surface expression, MLMECs and human U2OS cells expressing variant PCDH1 were seeded on fibronectin-coated glass coverslips 24 h pre-immunostaining. Cells were washed briefly in PBS before blocking with chilled PBS/10% FBS for 30 min at 4 °C. PCDH1 was detected by a human anti-EC1 mAb-3305 (5 µg/mL, Donnelly Centre and Department of Molecular Genetics, University of Toronto) or human anti-EC7 mAb-3677 (5 µg/mL, Donnelly Centre and Department of Molecular Genetics, University of Toronto) for 1 h at 4 °C. After washing with chilled PBS, the cells were fixed with 4% formaldehyde (Sigma-Aldrich) for 5 min followed by staining with an anti-human IgG Alexa Fluor™ 488 pAb (1:500 dilution, catalog No. A-11013, Thermo Fisher) or an anti-human IgG Alexa Fluor™ 555 antibody pAb (1:500 dilution, catalog No. A-21433, Thermo Fisher). For total PCDH1 expression, MLMECs transduced with Flag-tagged, human PCDH1(WT) or PCDH1(F83L) expressing retroviruses (see above) were plated on fibronectin-coated glass coverslips. Cells were fixed with 4% formaldehyde (Sigma) for 5 min and permeabilized with 0.1% Triton X-100 for 10 min at RT. After blocking, PCDH1 was detected by incubating cells with an anti-Flag mouse clone M2 mAb (1:500 dilution, catalog No. F1804, Sigma-Aldrich) followed by anti-mouse IgG Alexa Fluor™ 488 pAb (1:500 dilution, catalog No. A-11001, Thermo Fisher). All coverslips were mounted on glass slides using ProLong™ Gold Antifade Mountant containing DAPI (Thermo Fisher) and cells were examined using a Axio Observer Z1 wide-field epifluorescence microscope (Zeiss Inc., ZEN Imaging software, ZEN2 blue edition) with a 63x objective. Images were processed in Photoshop software (Adobe Systems, V.24.5.0).

## Animal welfare statement

Breeding, CRISPR/Cas9 genome engineering, and challenge studies with Syrian hamsters (*Mesocricetus auratus*) were conducted under IACUC-approved protocols in compliance with the Animal Welfare Act, PHS Policy, and other applicable federal statutes and regulations related to animals and experiments involving animals. The facilities where this research was conducted (Utah State University and USAMRIID) are accredited by the Association for Assessment and Accreditation of Laboratory Animal Care, International (AAALAC), and adhere to principles stated in the Guide for the Care and Use of Laboratory Animals, National Research Council, 2011. USAMRIID IACUC approved the protocols for the studies conducted at USAMRIID and Utah State University IACUC approved the protocols for the studies conducted at Utah State University.

## PCDH1-gene-edited Syrian hamsters

A panel of candidate sgRNAs was designed, assembled by overlapping PCR to generate human U6 promoter-driven sgRNA expression cassettes, and screened for genome-editing efficiency in BHK21 baby hamster kidney cells stably expressing Cas9. The best candidate sgRNA (sgRNA2: 5′-GACTACGGTTTTCCAGACTGGG-3′) targeted sequences encoding the homologs of human F83 and D85 in the hamster *PCDH1* gene (F79 and D81, respectively) (accession number NW_024429184.1). In addition, a knock-in, single donor strand (sequence: 5′-CCAAC ACCCTCATTGGGAGCCTTGCCGCTGAC          TACG GT**gcc**CA**aga**GTGG GTCATCTCTATAAACTAGAGGTAGGTGCTCCATATCTTC-3′ with the BaeGI cleavage site underlined and the bold, lowercase letters indicating the F83A and D85R mutation sites) was used for in vivo gene editing. The sgRNA was in vitro-transcribed and assembled into sgRNA/Cas9 ribonucleoprotein complexes, diluted with 10 mM RNase-free TE buffer to a concentration of 50 ng/µL sgRNA, 5 µM single donor strand, and 50 ng/µL Cas9, for pronuclear injections. PCDH1 gene-edited hamsters were produced following the procedure described in Jangra et al.[6]. Genomic DNA was isolated from hamster pups at the age of 2 weeks, a product flanking the sgRNA target sites was PCR-amplified and subjected to a T7 Endonuclease I assay (NEB) to detect indels. Amplicons from pups bearing indels were TOPO-cloned and sequenced to identify founder animals carrying nucleotide changes corresponding to the F83A and D85R mutations, in addition to the "10 amino acid" labeled founder animal, with a S76G mutation and a deletion of residues L77–D85 (corresponding to the homologous hamster residues S72 and L73-D81, respectively).

## Western blotting

PCDH1 expression in *PCDH1*-KI Syrian hamsters (described above) was confirmed by immunoblotting lung homogenates from WT and *PCDH1*-KI hamsters. Hamster lungs were placed in Cell Lysis Buffer (Invitrogen), containing 1 mM PMSF and protease inhibitor (Sigma-Aldrich), before homogenizing with zirconium beads. The supernatant was collected and the total amount of protein was determined via Bradford assay (BioRad). 30 µg of protein was added per lane and verified using a mouse anti-BetaActin clone 8H10D10 mAb (1:10,000 dilution in 1%NFDM, catalog No. MA5-15452, Thermo Fisher) along with an anti-mouse IgG IRDye-800CW pAb (1:10,000 dilution in 1%NFDM, catalog No. 926-32210, LI-COR). PCDH1 was detected using a pAb targeting the PCDH1 cytoplasmic tail (1:300 dilution in 1%NFDM, catalog No. PA5-83876, Thermo Fisher) by rocking for 1 h at RT, washing with PBS with 0.01% Tween 20 (PBST) and incubating with an anti-rabbit IgG Alexa Fluor™ Plus 800 pAb (1:10,000 dilution, catalog No. A32735, Thermo Fisher) for 1 h, rocking at RT. After thoroughly washing with PBST, western blots were imaged on a LI-COR Odyssey® Fc Imager (LI-COR Biosciences, LICOR Image Studio software, V.1.0.19) and the uncropped images are included in the Source Data file.

## Syrian hamster challenge studies

Groups of wild-type Syrian golden hamsters (*Mesocricetus auratus*) (Envigo) and *PCDH1*-KI (*PCDH1(10a.a.)* or *PCDH1(F83A/D85R)*) Syrian golden hamsters (Utah State University), 5-12 weeks old, male and female, were exposed to 2000 PFU of ANDV strain Chile-9717869 diluted in PBS, via the intranasal route and monitored for up to 35 days post-exposure. Animals were observed daily for clinical signs of disease, morbidity, and mortality. Moribund animals, described as being unresponsive or presenting with severe respiratory disease, were humanely euthanized on the basis of IACUC-approved criteria.

## Histopathology, immunohistochemistry, and in situ hybridization

Lungs harvested from *PCDH1(WT)* and *PCDH1(F83A/D85R)* hamsters (*n* = 3), 15 days post ANDV exposure, were fixed in buffered formalin for 30 days. Lung tissues were removed from biocontainment and processed at the USAMRIID histology lab. The tissues were trimmed, processed, embedded in paraffin, cut by microtomy, stained, coverslipped, and screened. For histopathology, 5-µm-thin sections were cut and stained with haematoxylin and eosin using standard procedures.

Immunohistochemistry was performed using the Dako Envision system (Dako Agilent Pathology Solutions). Briefly, after deparaffinization, peroxidase blocking, and antigen retrieval, sections were covered with an anti-SNV pAb (#1244, USAMRIID) that cross binds to Andes virus at a dilution of 1:5,000 and incubated at RT for 40 min. Sections were then rinsed, and a peroxidase-labeled polymer (secondary) was applied for 40 min. Slides were rinsed and a brown chromogenic substrate 3,3' Diaminobenzidine (DAB) solution (Dako Agilent Pathology Solutions) was applied for 8 min. After the substrate-chromogen solution was rinsed off, the slides were counterstained with hematoxylin and rinsed. The sections were dehydrated, cleared with Xyless, and then coverslipped.

In situ hybridization (ISH) was performed to detect ANDV RNA using the RNAscope 2.5 HD RED kit (Advanced Cell Diagnostics) according to the manufacturer's instructions. An ISH probe targeting ANDV S segment (GenBank accession number: NC_003466.1) was designed and synthesized by Advanced Cell Diagnostics (#900241, Advanced Cell Diagnostics). Tissue sections were deparaffinized with xylene, underwent a series of ethanol washes and peroxidase blocking, and heated in kit-provided, antigen retrieval buffer followed by digestion by kit-provided protease. Sections were exposed to ISH target probe pairs and incubated at 40 °C in a hybridization oven for 2 h. After rinsing, the ISH signal was amplified using a kit-provided Pre-amplifier and Amplifier conjugated to alkaline phosphatase and incubated with a Fast Red substrate solution for 10 min at RT. Sections were then counterstained with hematoxylin, air-dried, and cover-slipped.

### Serology of hamsters following ANDV challenge

ANDV Gn/Gc-specific IgG titers from infected Syrian hamster sera were determined by end-titer ELISA using rVSV-ANDV-Gn/Gc. Briefly, high-protein binding 96-well ELISA plates (Corning) were coated with 10 µg/ml rVSV-ANDV-Gn/Gc, diluted in PBS, and incubated overnight at 4 °C. Wells were then blocked in 5% milk protein in PBS/0.02% Tween 20 (2 h at RT). Serum samples were serially diluted in 5% milk protein in PBS/0.02% Tween 20 and added to antigen-coated plates (2 h at RT). Plates were washed with PBS/0.02% Tween 20 before adding HRP-conjugated goat anti-hamster IgG pAb (1:2,000 dilution, catalog No. 5220-0371, Seracare Life Sciences) (1 h at RT). Following a final wash, 2,2'-Azinobis [3-ethylbenzothiazoline-6-sulfonic acid]-diammonium salt (ABTS) substrate (Kirkegaard and Perry Laboratories, Inc.) was added and absorbance values were read at 405 nm using a Spectramax® plate reader (Molecular Devices, LLC).

### Statistics

The statistical parameters, including the nature of entity and exact value of $n$, deviations, $p$ values, and types of the statistical tests used, are reported in the figures and corresponding figure legends. The statistical analysis was carried out using Prism (GraphPad software, V.9)

### Reporting summary

Further information on research design is available in the Nature Portfolio Reporting Summary linked to this article.

## Data availability

The authors declare that the data and data sets supporting the findings of this study are available within the paper, its Supplementary Information files, and in the Figshare repository. The interfacial prediction data set generated in this study has been deposited in the Figshare database under accession code https://doi.org/10.6084/m9.figshare.23401916.v1[58], the PDB with the two modeled EC1 loop conformations has been deposited in the Figshare database under accession code https://doi.org/10.6084/m9.figshare.23398358[59]. The raw data generated in this study is provided in the Source Data file. Source data are provided with this paper.

## Code availability

The code used in this study to perform hierarchical clustering is available at https://github.com/chandranlab/pcdh1_interface or in the Figshare database under the accession code https://doi.org/10.6084/m9.figshare.23469236[56].

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

## Acknowledgements

The authors thank Estefania Valencia, Laura Polanco, Javier Janer, and Isabel Gutierrez for technical support, Eva Mittler for valuable input, and the Einstein Flow Cytometry Core. This work was supported by NIH grant AI132633 (to K.C., J.M.D., and Z.W.). R.K.J. was partly supported by NIH grant P20GM134974.

## Author contributions

Conceptualization, M.M.S., A.S.H., Z.W., R.K.J., and K.C.; Methodology, M.M.S., R.L., A.S.H., R.K.J.; Software, G.L.; Validation, M.M.S., R.L., A.S.H., A.I.K., R.R.B., S.R.M., Y.L., A.G., A.M.M., X.Z., R.K.J., P.G.-C.; Formal ana-lysis, M.M.S., A.S.H., G.L.; Investigation, M.M.S., R.L., A.S.H., A.I.K., R.R.B., S.R.M., Y.L., A.G., A.M.M., X.Z., R.K.J.; Resources, R.L., Y.L., F.A.R., P.G.-C.; Data curation, M.M.S., R.L., A.S.H., G.L., A.G., R.K.J.; Writing-original draft, M.M.S. and K.C.; Writing-review and editing, M.M.S., R.L., A.S.H., G.L., S.R.M., A.G., S.A., R.K.J., K.C.; Visualization, M.M.S., R.L., G.L., R.K.J., K.C.; Supervision, A.S.H., J.M.D., Z.W., K.C.; Project Administrator, K.C.; Funding Acquisition, K.C.

## Competing interests
