## [Peer Review File · Nature Communications]

Two point mutations in protocadherin-1 disrupt hantavirus recognition and afford protection against lethal infectionEditorial Note: This manuscript has been previously reviewed at another journal that is not operating a transparent peer review scheme. This document only contains reviewer comments and rebuttal letters for versions considered at Nature Communications.

Reviewers' Comments:

Reviewer #1:

Remarks to the Author:

This revised manuscript by Slough et al., is a follow up to a seminal study by Chandran et al., which defined PCDH1 as a receptor for ANDV and SNV. Prior studies by the same group with Ebola virus and NPC1 led to the identification of an entry factor. Using similar approaches, authors further explore sequence contributions and make an attempt to define evolutionary constraints in sequence space.

While significant effort is spent in the revision to rebut prior reviewer comments, the overall premise of the manuscript remains the same. This is an incremental and partial advance that lacks clarity and therefore, it does not advance our knowledge. The most impactful aspect is that their previous finding that PCDH1 is important for SNV/ANDV is reaffirmed. In addition, the authors conclude that they have defined a PCDH1 is a bona fide entry receptor. What is the basis for the receptor designation? Is signaling involved?

Overall, this revision doesn't move the needle in terms of conceptual advances or technical merit.

Reviewer #2:

Remarks to the Author:

I thank the authors for providing a response to the comments I had raised. Although their reply addressed most of the comments, there is still a minor concern about title of the manuscript. Authors concluded the title on the basis of only ANDV results not SNV but the Figure 1 and Figure 2 strongly highlighted that the "Single residue F83 in PCDH1 is a key determinant of SNV infection" that should also be addresses in the title or there should be any broad, non-residue specific title.

Reviewer #3:

Remarks to the Author:

All of my concerns have been addressed.

Reviewer #4:

None

Reviewer #5:

Remarks to the Author:

Here, Slough et al present a revised manuscript reporting novel viral factors that restrict hantavirus host range. This work is highly significant to the field, as it links molecular determinants of infection to pathogenesis in both cell culture and live animal models. The authors have sufficiently responded to my initial concerns with sound methodology.

We thank the reviewers for their final comments and have incorporated their concerns in the finalized version of the manuscript.

Reviewer #1:

This revised manuscript by Slough et al., is a follow up to a seminal study by Chandran et al., which defined PCDH1 as a receptor for ANDV and SNV. Prior studies by the same group with Ebola virus and NPC1 led to the identification of an entry factor. Using similar approaches, authors further explore sequence contributions and make an attempt to define evolutionary constraints in sequence space.

While significant effort is spent in the revision to rebut prior reviewer comments, the overall premise of the manuscript remains the same. This is an incremental and partial advance that lacks clarity and therefore, it does not advance our knowledge. The most impactful aspect is that their previous finding that PCDH1 is important for SNV/ANDV is reaffirmed. In addition, the authors conclude that they have defined a PCDH1 is a bona fide entry receptor. What is the basis for the receptor designation? Is signaling involved? Overall, this revision doesn't move the needle in terms of conceptual advances or technical merit.

We thank the reviewer for their feedback, but respectfully disagree with their conclusions about the merits of the manuscript. We believe that mapping the binding site of the viral glycoprotein on PCDH1, demonstrating that sequence variation at this site influences cellular host range, and showing that two point mutations engineered into PCDH1 in a lethal animal model greatly reduces virulence are important contributions to the understanding of hantavirus biology.

Reviewer #2:

I thank the authors for providing a response to the comments I had raised. Although their reply addressed most of the comments, there is still a minor concern about title of the manuscript. Authors concluded the title on the basis of only ANDV results not SNV but the Figure 1 and Figure 2 strongly highlighted that the "Single residue F83 in PCDH1 is a key determinant of SNV infection" that should also be addresses in the title or there should be any broad, non-residue specific title.

To keep within the 15-word limit we decided not to designate specific findings to each ANDV and SNV in the title of the manuscript. Instead, we switched 'ANDV' to 'hantaviruses' to make it broader as the two EC1 residues are recognized by both SNV and ANDV and are likely to be utilized by other HCPS-causing hantaviruses that depend on PCDH1 for entry, not just ANDV.

Reviewer #3:

All of my concerns have been addressed.

We thank the reviewer for their input.

Reviewer #5:

Here, Slough et al present a revised manuscript reporting novel viral factors that restrict hantavirus host range. This work is highly significant to the field, as it links molecular determinants of infection to pathogenesis in both cell culture and live animal models. The authors have sufficiently responded to my initial concerns with sound methodology.

We thank the reviewer for their input.